# The genetic basis for ecological adaptation of the Atlantic herring revealed by genome sequencing

Alvaro Martinez Barrio[1,2†], Sangeet Lamichhaney[1†], Guangyi Fan[3,4†], Nima Rafati[1†], Mats Pettersson[1], He Zhang[4,5], Jacques Dainat[1,6], Diana Ekman[7], Marc Höppner[1,6], Patric Jern[1], Marcel Martin[7], Björn Nystedt[2], Xin Liu[4], Wenbin Chen[4], Xinming Liang[4], Chengcheng Shi[4], Yuanyuan Fu[4,8], Kailong Ma[4], Xiao Zhan[4], Chungang Feng[1], Ulla Gustafson[9], Carl-Johan Rubin[1], Markus Sällman Almén[1], Martina Blass[10], Michele Casini[11], Arild Folkvord[12,13,14], Linda Laikre[15], Nils Ryman[15], Simon Ming-Yuen Lee[3], Xun Xu[4], Leif Andersson[1,9,16*]

[1]Science for Life Laboratory, Department of Medical Biochemistry and Microbiology, Uppsala University, Uppsala, Sweden; [2]Science for Life Laboratory, Department of Cell and Molecular Biology, Uppsala University, Uppsala, Sweden; [3]State Key Laboratory of Quality Research in Chinese Medicine, Institute of Chinese Medical Sciences, University of Macau, Macau, China; [4]BGI-Shenzhen, Shenzen, China; [5]College of Physics, Qingdao University, Qingdao, China; [6]Bioinformatics Infrastructure for Life Sciences, Uppsala University, Uppsala, Sweden; [7]Science for Life Laboratory, Department of Biochemistry and Biophysics, Stockholm University, Stockholm, Sweden; [8]School of Biological Science and Medical Engineering, Southeast University, Nanjing, China; [9]Department of Animal Breeding and Genetics, Swedish University of Agricultural Sciences, Uppsala, Sweden; [10]Department of Aquatic Resources, Institute of Coastal Research, Swedish University of Agricultural Sciences, Öregrund, Sweden; [11]Department of Aquatic Resources, Institute of Marine Research, Swedish University of Agricultural Sciences, Lysekil, Sweden; [12]Department of Biology, University of Bergen, Bergen, Norway; [13]Hjort Center of Marine Ecosystem Dynamics, Bergen, Norway; [14]Institute of Marine Research, Bergen, Norway; [15]Department of Zoology, Stockholm University, Stockholm, Sweden; [16]Department of Veterinary Integrative Biosciences, Texas A&M University, Texas, United States

*For correspondence: leif. andersson@imbim.uu.se

†These authors contributed equally to this work

Competing interests: The authors declare that no competing interests exist.

**Abstract** Ecological adaptation is of major relevance to speciation and sustainable population management, but the underlying genetic factors are typically hard to study in natural populations due to genetic differentiation caused by natural selection being confounded with genetic drift in subdivided populations. Here, we use whole genome population sequencing of Atlantic and Baltic herring to reveal the underlying genetic architecture at an unprecedented detailed resolution for both adaptation to a new niche environment and timing of reproduction. We identify almost 500 independent loci associated with a recent niche expansion from marine (Atlantic Ocean) to brackish waters (Baltic Sea), and more than 100 independent loci showing genetic differentiation between spring- and autumn-spawning populations irrespective of geographic origin. Our results show that

both coding and non-coding changes contribute to adaptation. Haplotype blocks, often spanning multiple genes and maintained by selection, are associated with genetic differentiation.

The Atlantic herring (*Clupea harengus*) is a pelagic fish that occurs in huge schools, up to billions of individuals. The herring fishery has been crucial for food security and economic development in Northern Europe and currently ranks among the five largest fisheries in the world with nearly 2 million tons fish landed annually (*FAO, 2014*). The herring is one of few marine fishes that reproduce throughout the Baltic Sea where the salinity drops to 2–3‰ in the Bothnian Bay, compared with 35‰ in the Atlantic Ocean (*Figure 1A*). This ecological adaptation must be recent because the brackish Baltic Sea has only existed for 10,000 years following the last glaciation (*Andrén et al., 2011*). Fishery biologists have for more than a century recognized stocks of herring defined by spawning location, spawning time, morphological characters and life history parameters (*Iles and Sinclair, 1982*; *McQuinn, 1997*). Several decades of genetic studies based on limited numbers of genetic markers (allozymes, microsatellites or SNPs) have not been able to verify this divergence; extremely low levels of differentiation even between geographically distant populations as well as between spring- and autumn-spawning herring have been observed (*Andersson et al., 1981*; *Ryman et al., 1984*; *Larsson et al., 2007*; *2010*, *Limborg et al., 2012*). It has been proposed that lack of precision in homing behaviour of herring causes sufficient gene flow between stocks to counteract genetic differentiation (*McQuinn, 1997*). However, in a recent study we constructed an exome assembly and used this in combination with whole genome sequencing of eight population samples and found more than 400,000 SNPs (*Lamichhaney et al., 2012*). We confirmed lack of differentiation at most loci, whereas a small percentage showed highly significant differentiation. Simulations demonstrated that the distribution of fixation index ($F_{ST}$)-values among herring populations deviated significantly from expectation for selectively neutral loci.

Genetic studies of ecological adaptation in natural populations is challenging because genetic differentiation caused by natural selection is often confounded with genetic differences due to genetic drift caused by restricted effective population sizes. An ideal species for studying the genetic basis of ecological adaptation should comprise subpopulations of infinite size and exposed to different ecological conditions. In such a species there is minute genetic drift and genetic differentiation is caused by selection resulting in local adaptation. The herring is close to being such an ideal subject for studies of ecological adaptation due to the extremely low levels of genetic differentiation at most loci as documented in previous studies (*Andersson et al., 1981*; *Ryman et al., 1984*; *Larsson et al., 2007*; *2010*; *Limborg et al., 2012*; *Lamichhaney et al., 2012*). This unique opportunity together with herring being such a valuable natural resource prompted us to generate a genome assembly and perform genome sequencing of populations adapted to different ecological conditions.

Here we present a high-quality genome assembly for the Atlantic herring, and results of whole genome sequencing of 20 population samples using pooled DNA. The results were verified by individual genotyping using a custom-made 70k SNP array. Our study addresses two fundamentally different types of adaptations; one example of niche expansion (adaptation to low salinity), and one example of sympatric balancing selection (variation in the timing of reproduction). The results provide a comprehensive list of hundreds of independent loci underlying ecological adaptation and shed light on the relative importance of coding and non-coding variation. The results have important implications for sustainable fishery management, and provide a road map for cost effective high-resolution characterization of genetic diversity in natural populations.

## Results

### Genome assembly and annotation

Clupeiformes represents an early diverging clade of the otomorpha (*Near et al., 2012*) (*Figure 2A*). The genome size for herring has been estimated at ~850 Mb (*Hinegardner and Rosen, 1972*; *Ida et al., 1991*; *Ohno et al., 1969*) with no recent whole genome duplications reported. We performed whole genome assembly based on short read sequencing of libraries ranging from 170 bp to

**eLife digest** The Atlantic herring is one of the most common fish in the world and has been a crucial food resource in northern Europe. One school of herring may comprise billions of fish, but previous studies had only revealed very few genetic differences in herring from different geographic regions. This was unexpected since Atlantic herring is one of the few marine species that can reproduce throughout the brackish Baltic Sea, which can be about a tenth as salty as the Atlantic Ocean.

This unexpected finding could be explained in at least two different ways. Firstly, perhaps Atlantic herring are flexible enough to adapt to very different environments (i.e. high or low salinity) without much genetic change. Secondly, the previous studies only looked at a handful of sites in the Atlantic herring's genome and so it is possible that genetic differences at other genes control this fish's adaptation instead.

Now, Martinez Barrio, Lamichhaney, Fan, Rafati et al. have sequenced entire genomes from groups of Atlantic herring and revealed hundreds of sites that are associated with adaptation to the Baltic Sea. The analysis also identified a number of genes that control when these fish reproduce by comparing herring that spawn in the autumn with those that spawn in spring. This is important because natural populations must carefully time when they reproduce to maximize the survival of their young.

These new findings provide compelling evidence that changes in protein-coding genes and stretches of DNA that regulate the expression of other genes both contribute to adaptation in herrings. The analysis also clearly shows that variants of genes that contribute to adaptation were likely to evolve over time by accumulating multiple sequence changes affecting the same gene. Furthermore, these gene variants essentially form a rich "tool-box" that underlies the Atlantic herring's adaptation to its environment, and different subpopulations of herring were found to have their own optimal sets of gene variants. For instance, autumn-spawning herring and spring-spawning herring from the Baltic Sea both have gene variants that favor adaptation to low salinity. However, autumn-spawning Baltic herring also share gene variants that favor spawning in the autumn with autumn-spawning herring from the North Sea, but not with spring-spawning Baltic herring.

The next step will be to study how the 500 or so genes identified affect adaptation at the molecular level. This will likely involve experiments with other model fish such as zebrafish and sticklebacks. Finally, these new findings can be directly applied to monitor stocks of herring to make herring fisheries more sustainable.

20 kb insert sizes (*Supplementary file 1A*). The 808 Mb assembly had a scaffold N50 of 1.84 Mb with 23,336 predicted coding gene models. It showed a high degree of completeness based on RNAseq alignments, core gene analyses and comparisons to other fish gene sets (*Table 1*, *Supplementary files 2*, *3A–D*, *Figure 2B*, *Figure 2—figure supplements 1–2*). The GC content was 44%, and repetitive elements made up 31% of the assembly (*Table 1*). Alignments of synthetic long reads (SLRs; Illumina) failed to significantly improve the assembly due to coincidental gaps between the assembly and the SLRs, but proved useful in phasing parental alleles (Materials and methods; *Figure 2—figure supplements 3–4*) and dramatically improved the discovery of indels larger than 30 bp compared to short Illumina reads (*Supplementary file 1F*). We identified 150 endogenous retroviruses (ERVs) constituting ~0.14% of the genomic sequence but none included open reading frames in all *gag*, *pol* and *env* genes (*Supplementary file 1*, *Figure 2—figure supplement 5*).

## Population genetics and demographic history

Whole genome pooled sequencing was done using 20 population samples of herring from the Baltic Sea, Skagerrak, Kattegat, North Sea, Atlantic Ocean and Pacific Ocean (*Figure 1A*; *Table 2*); the latter sample represents the closely related Pacific herring (*Clupea pallasii*). Each pool comprised 47–100 fish and was sequenced to ~30x coverage. Furthermore, 16 fish, eight Baltic and eight Atlantic herring (*Table 2*), were sequenced individually to ~10x coverage. All data were aligned to the

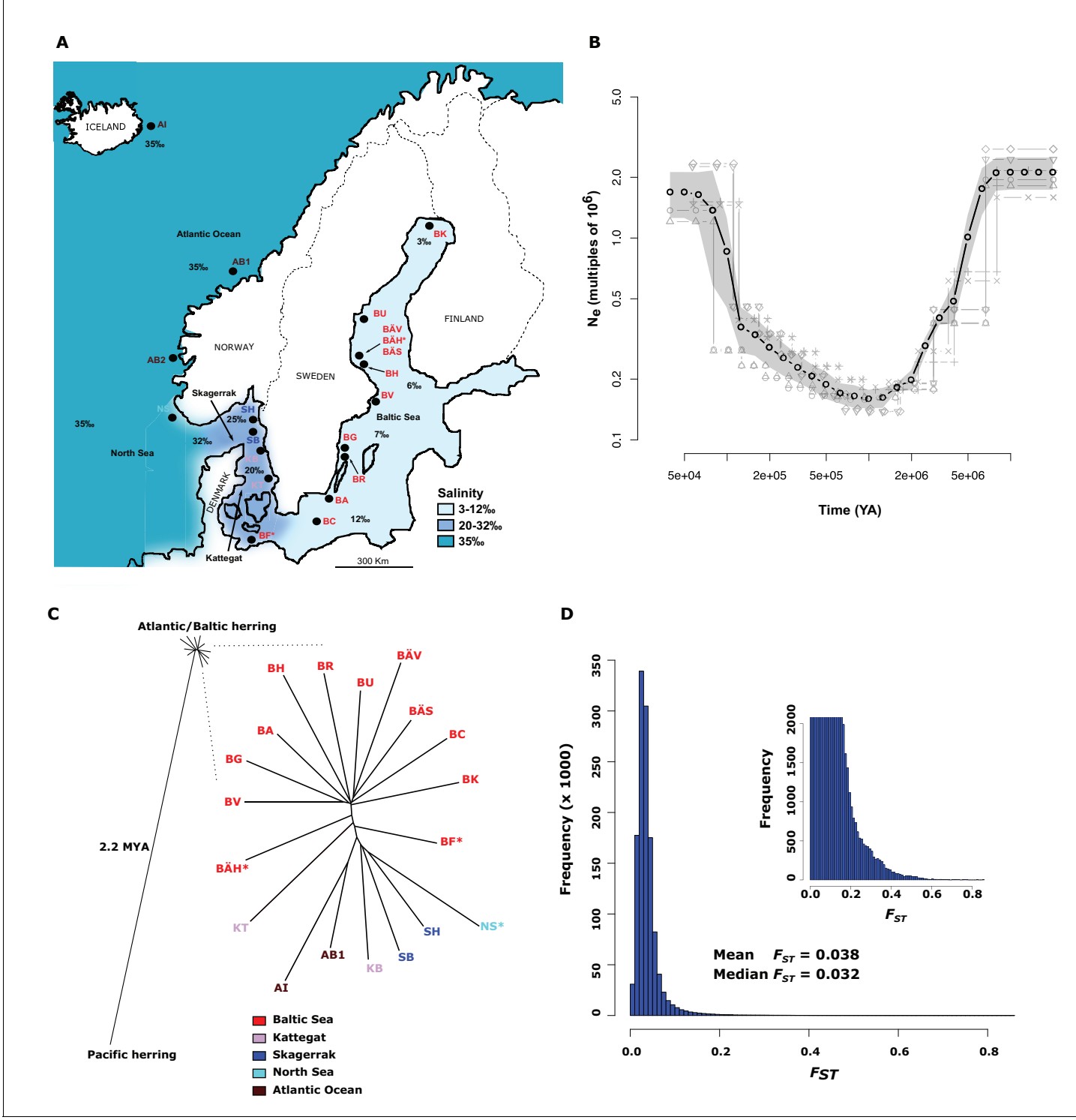

**Figure 1.** Demographic history and phylogeny. (**A**) Geographic location of samples. The salinity of the surface water in different areas is indicated schematically. Autumn spawners are marked with an asterisk. (**B**) Demographic history. Black circles indicate effective population size over time estimated by diCal (*Sheehan et al., 2013*); estimates are averages from four arbitrarily chosen genomic regions. The grey field is confidence interval ( ± 2 sd), while light grey lines show the underlying estimates from each genomic region. (**C**) Neighbor-joining phylogenetic tree. The evolutionary distance between Atlantic and Pacific herring was calculated using mtDNA cytochrome B sequences; right panel, zoom-in on the cluster of Atlantic and Baltic herring populations. Colour codes for sampling locations are the same as in *Figure 1A*. (**D**) Global distribution of $F_{ST}$–values based on 19 populations of Atlantic and Baltic herring. The inset illustrates the tail of the distribution. The mean and median of this distribution are indicated. To reduce the $F_{ST}$ sampling variance, we only used SNPs with ≥30x coverage in each population.

*Figure 1 continued on next page*

*Figure 1 continued*

The following figure supplement is available for figure 1:

**Figure supplement 1.** Population genetics and Q-Q plot.

reference assembly and SNPs were called after rigorous quality filtering. We found 8.83 million SNPs when Pacific herring was included and 6.04 million among Atlantic and Baltic herring.

Average nucleotide diversity was estimated by counting the frequency of heterozygous sites in the reference individual after stringent filtering for sequence quality and coverage (within one standard deviation of mean coverage). The estimate was one heterozygous site per 309 bp, giving a nucleotide diversity of 0.32%; no estimate based on the 16 herring sequenced individually deviated significantly from this value and there was no significant difference between Atlantic and Baltic herring. The average decay of linkage disequilibrium between loci was very steep, with average $r^2$ falling to 0.1 at a distance of 100 base pairs (*Figure 1—figure supplement 1A*).

The allele frequency distribution deviated significantly from the one expected for selectively neutral alleles at genetic equilibrium ($p < 2 \times 10^{-16}$, Kolmogorov-Smirnov test), due to an excess of rare alleles (*Figure 1—figure supplement 1B*) consistent with population expansion. The result is supported by the genome-wide distribution of Tajima's D, which shows a global shift towards negative values (mean $= -0.57 \pm 0.01$; *Figure 1—figure supplement 1C*). A demographic analysis using the diCal software (*Sheehan et al., 2013*) confirmed that herring have experienced an expansion in effective population size, roughly five- to ten-fold, and that the current $N_e$ is on the order of $10^6$ individuals (*Figure 1B*); the results for Baltic and Atlantic herring were essentially identical. The result indicates that the effective population size minimum occurred at around one to two MYA, after the onset of the Quaternary ice age.

## Phylogeny

The neighbor-joining phylogenetic tree including Atlantic, Baltic and Pacific herring shows a large phylogenetic distance between Pacific and Atlantic herring, as compared with the tiny genetic divergence among samples of Atlantic and Baltic herring (*Figure 1C*). We estimated the split between Atlantic and Pacific herring to ~2.2 million years ago based on mtDNA cytochrome B sequence divergence. The phylogenetic tree is consistent with minute differentiation at selectively neutral loci in Atlantic herring (*Ryman et al., 1984*; *Lamichhaney et al., 2012*); all subpopulations in the Eastern North Atlantic may have expanded from a common ancestral population after the last glaciation as indicated by demographic analysis (*Figure 1B*).

A closer examination of the tight cluster of Atlantic and Baltic herring populations reveals some structure consistent with geographic origin (*Figure 1C*). Samples from the Baltic Sea cluster on one half while samples from marine waters cluster on the other half of the tree. Only three populations are located at intermediate positions. Two of these are autumn-spawners from the Baltic Sea (BÄH and BF), indicating that autumn-spawning herring are genetically distinct from spring- and summer-spawning herring. The third sample (KT) at an intermediate position was sampled outside the spawning season and at the border between Kattegat and Baltic Sea and may represent a mixed sample of local Kattegat population and fish that spawn in the Baltic Sea but migrate into Kattegat for feeding.

## Genetic adaptation to a new niche environment

The Atlantic (*Clupea harengus harengus*) and Baltic herring (*Clupea harengus membras*) were classified as subspecies by *Linnaeus (1761)* in the 18th century. They are adapted to strikingly different environments, in particular regarding salinity that ranges from 2–3‰ in the Gulf of Bothnia to 12‰ in Southern Baltic Sea, whereas salinity in Kattegat, Skagerrak, North Sea and Atlantic Ocean is in the range 20‰–35‰ (*Figure 1A*; *Table 2*). To reveal loci underlying genetic adaptation associated

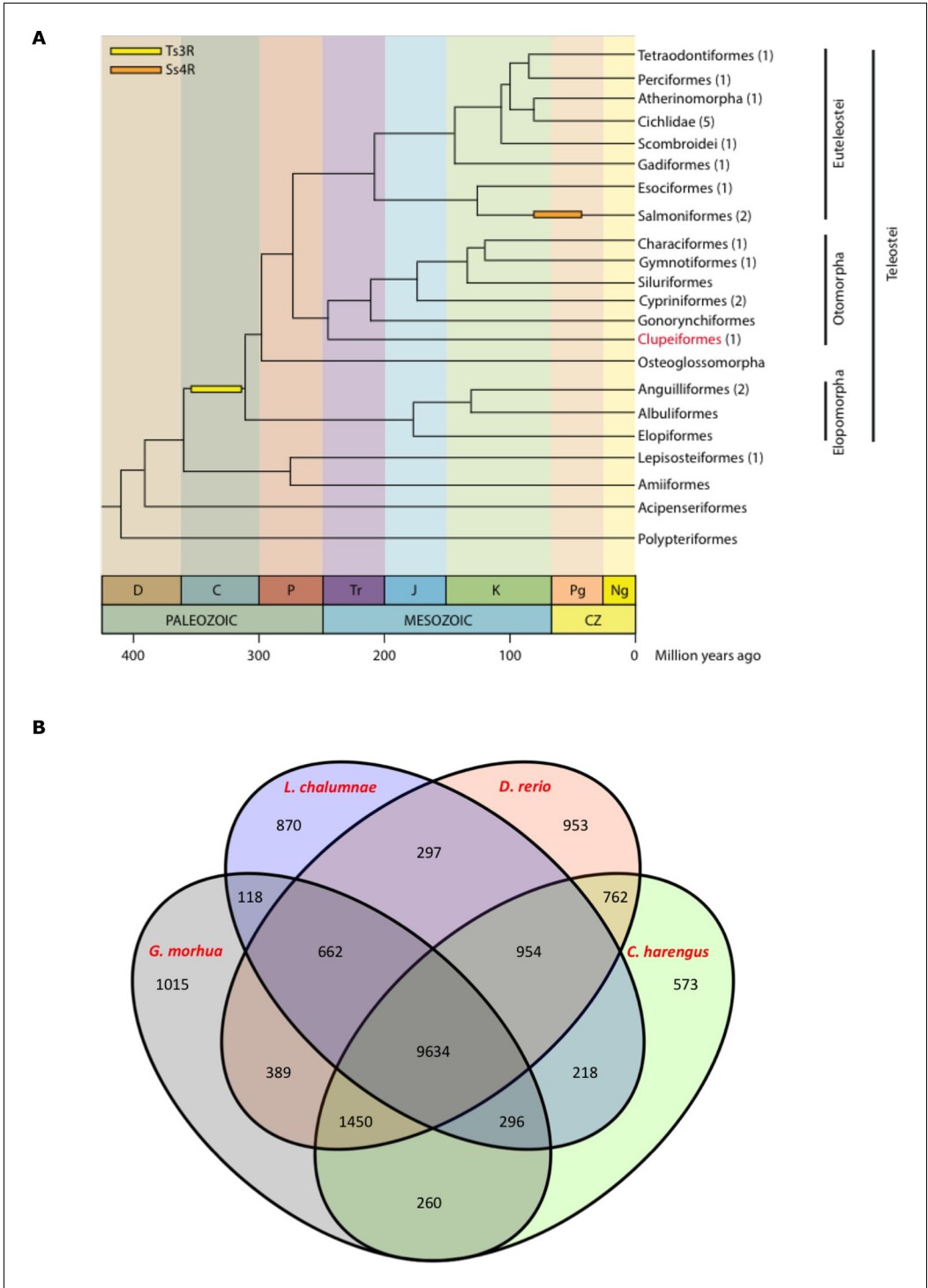

**Figure 2.** Genome assembly and annotation. (**A**) Phylogeny of ray-finned fishes (*Actinopterygii*) from the Devonian to the present, time-calibrated to the geological time scale based on *Near et al. (2012)*. Geological abbreviations: C (Carboniferous), CZ (Cenozoic), D (Devonian), J (Jurassic), K (Cretaceous), Ng (Neogene), P (Permian), Pg (paleogene) and Tr (Triassic). Dating of the specific rounds of whole genome duplication is based on *Glasauer and Neuhauss (2014)*. Abbreviations: Ts3R (teleost-specific third round) and Ss4R (salmonid-specific fourth round) of duplication. The number of species with a genome assembly available is marked within parentheses after their group's name. Atlantic herring belongs to Clupeiformes, the order indicated in red letters. (**B**) Orthologous gene families across four fish genomes (*C. harengus, D. rerio, L. chalumnae* and *G. morhua*).

The following figure supplements are available for figure 2:

**Figure supplement 1.** Schematic overview of the annotation pipeline.

*Figure 2 continued on next page*

*Figure 2 continued*

**Figure supplement 2.** Density plot of the Annotation Edit Distance (AED) score distribution for gene builds rc4 and rc5.

**Figure supplement 3.** Overall read length histogram for the five synthetic long reads (SLR) libraries.

**Figure supplement 4.** Read coverage of the assembly with synthetic long reads (SLRs) is uneven and not Poisson-shaped.

**Figure supplement 5.** Phylogeny of endogenous retroviruses (ERVs).

with the recent niche expansion into brackish waters after the last glaciation we compared allele frequencies, SNP by SNP, in two superpools: one Atlantic including all populations from Atlantic Ocean, Skagerrak and Kattegat and a pool comprising all samples collected in Baltic Sea; this is justified by low differentiation at neutral loci as documented by the low $F_{ST}$-values when comparing all samples of Atlantic and Baltic herring (*Figure 1D*). Samples of autumn-spawning herring, a possible confounding factor, were excluded from the analysis. We used a stringent significance threshold of $p<1\times10^{-10}$ (Bonferroni correction, $p=8.2\times10^{-9}$).

We identified 46,045 SNPs that showed an allele frequency difference with $p<1\times10^{-10}$ in the $\chi^2$ test (*Figure 3A*; *Supplementary file 3A*). An important question is how many independent loci these represent. A conservative estimate of 472 independent loci was obtained (i) by only using SNPs with $p<1\times10^{-20}$, (ii) by taking into account gaps in the assembly and (iii) by using the Comb-P software (*Pedersen et al., 2012*) to combine strongly correlated SNPs from the same genomic region (see Materials and methods). *Figure 3A* (lower panel) illustrates one of the most striking associations. For a large part of scaffold 218 there are no significant differences among Atlantic and Baltic samples whereas there are striking allele frequency differences over a 119.4 kb region; this is a characteristic pattern for differentiated regions, indicating that genetic adaptation typically occur as large haplotype blocks, often including multiple genes. A phylogenetic tree based on SNPs showing genetic differentiation between Atlantic and Baltic (*Figure 3B*) differs profoundly from the tree

**Table 1.** Summary of the herring assembly compared to other sequenced fish genomes.

| Species | Herring (*Clupea harengus*) | Zebrafish (*Danio rerio*) | Cod (*Gadus morhua*) | Coelacanth (*Latimeria chalumnae*) | Stickleback (*Gasteosteus aculeatus*) |
|---|---|---|---|---|---|
| Estimated genome size (Mb) | 850 | 1,454[a] | 830[b] | 3,530[c] | 530[d] |
| Assembly size (Mb) | 808 | 1,412 | 753[b] | 2,861[e] | 463[f] |
| Contig N50 (kb) | 21.3 | 25.0 | 2.8 | 12.7 | 83.2 |
| Scaffold N50 (Mb) | 1.84 | 1.55 | 0.69 | 0.92 | 10.8 |
| Sequencing technology[g] | I | S+I | R+I | I | S |
| Repeat content | 30.9 | 52.2 | 25.4 | 27.7 | 25.2 |
| %GC content | 44.1 | 36.7 | 45.4 | 43.0 | 44.6 |
| Heterozygosity | 1/309 | n.a. | 1/500 | 1/435 | 1/700 |
| Protein-coding gene count | 23,336 | 26,459 | 22,154 | 19,033 | 20,787 |

[a](*Freeman et al., 2007*; *Vinogradov, 1998*; *Howe et al., 2013*)

[b](*Star et al., 2011*)

[c]Genome size calculated as pg x 0.978 × $10^9$ bp/pg; picogram values taken from *Cimino and Bahr (1974)*

[d](*Vinogradov, 1998*; *Jones et al., 2012*)

[e](*Amemiya et al., 2013*)

[f](*Jones et al., 2012*)

[g]I=Illumina sequencing; S=Sanger sequencing; R=Roche 454 n.a.=not available

**Table 2.** Samples of herring used for whole genome resequencing.

| Locality[a] | Sample | n | Position | | Salinity (‰) | Date (yy/mm/dd) | Spawning season |
|---|---|---|---|---|---|---|---|
| **Baltic Sea** | | | | | | | |
| Gulf of Bothnia (Kalix)[b] | BK | 47 | N 65°52′ | E 22°43′ | 3 | 800629 | spring |
| Bothnian Sea (Hudiksvall) | BU | 100 | N 61°45′ | E 17°30′ | 6 | 120419 | spring |
| Bothnian Sea (Gävle) | BÄV | 100 | N 60°43′ | E 17°18′ | 6 | 120507 | spring |
| Bothnian Sea (Gävle) | BÄS | 100 | N 60°43′ | E 17°18′ | 6 | 120718 | summer |
| Bothnian Sea (Gävle) | BÄH | 100 | N 60°44′ | E 17°35′ | 6 | 120904 | autumn |
| Bothnian Sea (Hästskär)[c] | BH | 50 | N 60°35′ | E 17°48′ | 6 | 130522 | spring |
| Central Baltic Sea (Vaxholm)[b] | BV | 50 | N 59°26′ | E 18°18′ | 6 | 790827 | spring |
| Central Baltic Sea (Gamleby)[b] | BG | 49 | N 57°50′ | E 16°27′ | 7 | 790820 | spring |
| Central Baltic Sea (Kalmar) | BR | 100 | N 57°39′ | E 17°07′ | 7 | 120509 | spring |
| Central Baltic Sea (Karlskrona) | BA | 100 | N 56°10′ | E 15°33′ | 7 | 120530 | spring |
| Central Baltic Sea | BC | 100 | N 55°24′ | E 15°51′ | 8 | 111018 | unknown |
| Southern Baltic Sea (Fehmarn)[b] | BF | 50 | N 54°50′ | E 11°30′ | 12 | 790923 | autumn |
| **Kattegat, Skagerrak, North Sea, Atlantic Ocean** | | | | | | | |
| Kattegat (Träslövsläge)[b] | KT | 50 | N 57°03′ | E 12°11′ | 20 | 781023 | unknown |
| Kattegat (Björköfjorden) | KB | 100 | N 57°43′ | E 11°42′ | 23 | 120312 | spring |
| Skagerrak (Brofjorden) | SB | 100 | N 58°19′ | E 11°21′ | 25 | 120320 | spring |
| Skagerrak (Hamburgsund)[b] | SH | 49 | N 58°30′ | E 11°13′ | 25 | 790319 | spring |
| North Sea[b] | NS | 49 | N 58°06′ | E 06°10′ | 35 | 790805 | autumn |
| Atlantic Ocean (Bergen)[b] | AB1 | 49 | N 64°52′ | E 10°15′ | 35 | 800207 | spring |
| Atlantic Ocean (Bergen)[c] | AB2 | 8 | N 60°35′ | E 05°00′ | 33 | 130522 | spring |
| Atlantic Ocean (Höfn) | AI | 100 | N 65°49′ | W 12°58′ | 35 | 110915 | spring |
| **Pacific Ocean** | | | | | | | |
| Strait of Georgia (Vancouver) | PH | 50 | - | - | 35 | 121124 | - |

[a]Places where the sample was landed (if known) are given in parenthesis
[b]Samples from previous study (**Lamichhaney et al., 2012**)
[c]Eight Baltic herring from the BH sample and eight Atlantic herring from the AB2 sample were used for individual sequencing n=number of fish

based on all SNPs (*Figure 1C*). With the exception of the two autumn-spawning populations BF and BÄH from the Baltic Sea, the position of all other populations match the variation in salinity perfectly with the population samples from the North Sea and Atlantic Ocean (35‰) at one end of the tree and samples from the brackish Baltic Sea (3‰–12‰) at the other end and with samples from Skagerrak (25‰) and Kattegat (20‰) at intermediate positions. The low genetic differentiation among Baltic samples, excluding the two autumn-spawning populations BF and BÄH, suggests that adaptation to brackish waters is a derived state.

*Figure 3C* (upper panel) shows estimated allele frequencies for highly differentiated SNPs from five genomic regions in six population samples, each region showing an underlying genetic architecture with large and distinctly defined haplotype blocks. The Atlantic Ocean and North Sea samples are both nearly fixed for the reference allele at these SNPs. In contrast, the samples of Baltic herring were close to fixation for the alternate alleles. Interestingly, the sample (SB) collected in Skagerrak (salinity ~25‰) is most similar to the Atlantic Ocean and North Sea samples, but consistently shows a trend towards more intermediate allele frequencies at these loci.

We developed a 70k custom SNP chip to study differentiated regions in more detail and to use data from individual fish to confirm associations detected by pooled sequencing. The chip included 13,355 neutral SNPs evenly distributed across the genome and 59,205 SNPs showing genetic differentiation between subpopulations. Thirty fish each from 12 populations were used in the SNP

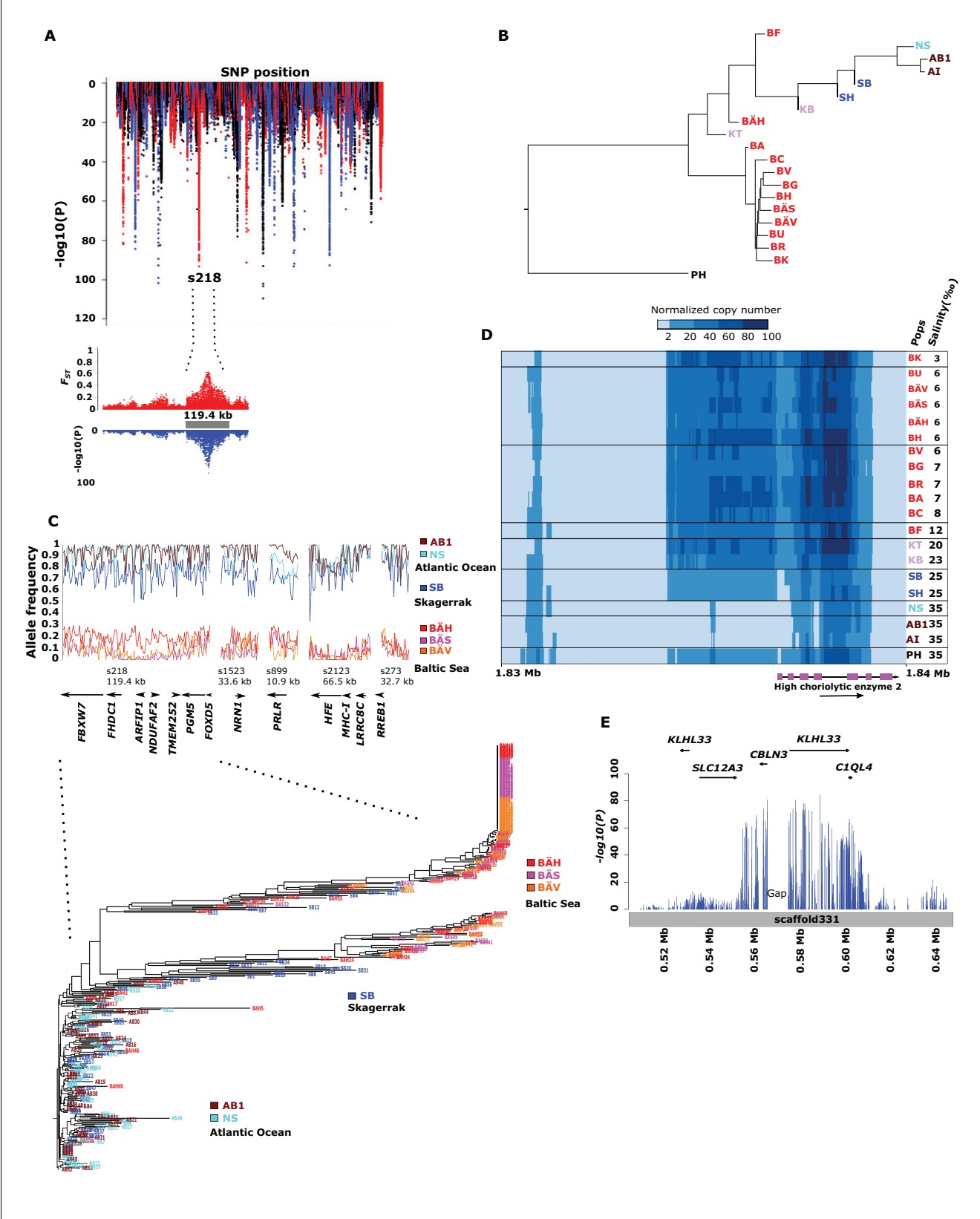

**Figure 3.** Genetic differentiation between Atlantic and Baltic herring. (**A**) Manhattan plot of significance values testing for allele frequency differences between pools of herring from marine waters (Kattegat, Skagerrak, Atlantic Ocean) versus the brackish Baltic Sea. Lower panel, corresponding plot for
*Figure 3 continued on next page*

*Figure 3 continued*

scaffold 218 only; both *P*- and $F_{ST}$–values are shown. (**B**) Neighbor-joining phylogenetic tree based on all SNPs showing genetic differentiation in this comparison ($p<10^{-10}$). (**C**) Comparison of allele frequencies in five strongly differentiated regions. The major allele in the AB1 sample (Atlantic Ocean) was used as reference at each SNP. Lower panel, neighbor-joining tree based on haplotypes formed by 128 differentiated SNPs from scaffold 218. (**D**) Heat map showing copy number variation partially overlapping the *HCE* gene. Orientation of transcription is marked with an arrow; the position of SNPs significant in the $\chi^2$ test is indicated by stars. Population samples and salinity at sampling locations are indicated to the right; abbreviations are explained in *Table 2*. (**E**) Strong genetic differentiation between Atlantic and Baltic herring in a region downstream of *SLC12A3*; statistical significance based on the $\chi^2$ test is indicated.

The following figure supplements are available for figure 3:

**Figure supplement 1.** Comparison of allele frequencies estimated using pooled whole genome sequencing or by individual genotyping using a SNP chip.

**Figure supplement 2.** Additional neighbor-joining trees for the contrast Atlantic versus Baltic.

screen. There was an excellent correlation between allele frequencies estimated with pooled sequencing and with the SNP chip (*Figure 3—figure supplement 1*). We constructed a phylogenetic tree (*Figure 3C*, lower panel) for haplotypes of highly differentiated SNPs from scaffold 218 present among individual fish from six representative populations, after phasing haplotypes using BEAGLE (*Browning and Browning, 2007*). As expected all fish from Atlantic Ocean and North Sea carried closely related "Atlantic" haplotypes. Two major haplotype groups were present among Baltic herring and with few exceptions Baltic herring carried only "Baltic" haplotypes. Fish from Skagerrak predominantly carried Atlantic haplotypes, but with a considerable proportion of Baltic haplotypes. Phylogenetic trees for other top scaffolds are presented in *Figure 3—figure supplement 2*.

There are many environmental and ecological differences between Atlantic Ocean and Baltic Sea e.g. temperature variability, eutrophication of the Baltic Sea, zooplankton and predator populations), but the most obvious difference concerns salinity. We used the Bayenv 2.0 (*Günther and Coop, 2013*) software to reveal which of the 472 independent loci detected with the $\chi^2$ test showed the most consistent correlation with salinity. This analysis identified 3,335 SNPs from 122 independent regions with highly significant association to salinity (*Supplementary file 3A*). Twenty-one of the genes in these regions have previously been associated with hypertension in human and 36 of these genes showed differential expression in sticklebacks kept in freshwater or sea water (*Supplementary file 3A*).

Here we present three loci with striking association to salinity. Firstly, the 11 kb region in scaffold 899 (*Figure 3C*) contains a single gene, *prolactin receptor (PRLR)*, that is essential for mammalian reproduction but has a central role for osmoregulation in fish (*Manzon, 2002*), and possibly in mammals (*Schennink et al., 2015*). Secondly, strong genetic differentiation was also observed at scaffold 346 (*Figure 3A*; $p<1\times10^{-39}$). This signal overlaps *HCE* encoding high choriolytic enzyme. This locus was also identified as one of the most differentiated region in our screen for structural changes (*Supplementary file 3B*). A 4 kb region including part of the coding sequence showed a massive copy number amplification that had a strong negative correlation with salinity (*Figure 3D*). The outgroup, Pacific herring, showed an intermediate copy number. Interestingly, the Pacific herring spawns exclusively in shallow nearshore waters (*Hay et al., 2009*) often in estuaries and tidal zones where salinity varies, in contrast to deeper-spawning Atlantic herring. HCE is a protease, also denoted hatching enzyme, that solubilizes the inner layer of the egg envelope during hatching and adaptive evolution of this protein in relation to salinity has been reported (*Kawaguchi et al., 2013*). In herring, we found no coding changes implying altered transcriptional regulation. In fact, massive amplification of the promoter region is expected to alter gene expression. Hatching of the egg is probably a particularly challenging stage of development for a marine fish adapting to brackish conditions. Thirdly, a ∼65 kb region downstream of *solute carrier family 12 (sodium/chloride transporter) member 3 (SLC12A3)* shows strong correlation with salinity (*Figure 3E*, *Supplementary file 3A*). *SLC12A3,* which has an established role in regulating osmotic balance, is associated with hypertension in human and shows differential expression in kidney tissue between sticklebacks kept in freshwater or sea water (*Wang et al., 2014*).

## Genetic basis underlying timing of reproduction

Herring spawn from early spring to late fall. Prior to this study it was unknown if spawning time is entirely due to phenotypic plasticity, set by nutritional status and environmental conditions, or if genetic factors contribute (*McQuinn, 1997*). For example, it has been hypothesized that spawning time in the Baltic Sea is regulated by productivity of the system affecting maturation of fish prior to spawning (*Aneer, 1985*). To study this important question we collected spawning herring from the same geographic area, close to Gävle (Sweden), in May, July and September (*Table 2*). Our sampling included two other autumn-spawning populations collected in 1979, one from North Sea and the other from Southern Baltic Sea. We formed two superpools including three autumn-spawning and 10 spring-spawning population samples, respectively; the summer-spawners and one population of non-spawning herring (KT in *Table 2*) were excluded from the initial analysis. We identified 10,195 SNPs with significant allele frequency differences between pools ($p<1\times10^{-10}$) and 69 regions with copy number variation ($p<0.001$) (*Figure 4A*); the highly differentiated SNPs represented at least 125 independent loci based on our strict criteria (see Materials and methods). The result demonstrates for the first time that autumn- and spring-spawning herring are genetically distinct and indicates that genetic factors affect spawning time. In a phylogenetic tree based on these 10,195 SNPs the autumn-spawning populations from the Baltic Sea and North Sea tended to cluster with spring-spawning herring from the Atlantic Ocean (*Figure 4B*).

A general linear mixed model was used to identify which of the 125 independent loci showed the most consistent allele frequency differences between spring and autumn spawners. This analysis revealed 17 independent genomic regions that passed the stringent significance threshold of $p<10^{-10}$ (Bonferroni correction, $p=4.9\times10^{-6}$) (*Supplementary file 3C*). We then illustrate the striking allele frequency differences at the four most significant regions using data from six different populations. As observed for the genetic adaptation to declined salinity (above), the most significant regions underlying seasonal reproductive timing typically consists of large haplotype blocks often containing multiple genes. Spring-spawning Atlantic and Baltic herring showed nearly identical allele frequencies at these loci while autumn-spawning herring from Baltic Sea and North Sea showed high frequencies of the alternate alleles (*Figure 4C*). Remarkably, summer-spawning herring showed a clear trend towards intermediate allele frequencies at all loci, most pronounced for scaffold 481 (*Figure 4C*). This may either reflect that this sample is an admixture of spring- and autumn-spawning herring or that it represents a distinct population. To explore this we investigated deviations from Hardy-Weinberg equilibrium using the $F_{IT}$ statistics because we expect a heterozygote deficiency if this is a mix of two populations. The results, based on 1,500 SNPs all showing strong genetic differentiation between spring- and autumn-spawners and genotyped individually using the SNP chip, showed that the summer spawners (BÄS) did not deviate markedly from $F_{IT} = 0$ and in fact to a lesser extent than the spring-spawning population (BÄV) sampled at the same locality (*Figure 4—figure supplement 1*). For instance, individual genotyping of the highly differentiated SNPs from scaffold 481 (*Figure 4C*) resulted in mean $F_{IT} = -0.10$ (excess of heterozygotes) for the summer spawners (BÄS) whereas if the sample had constituted an equal mix of spring- and autumn spawners from the same locality (BÄV and BÄH) the expected $F_{IT}$-value would have been 0.46 (strong heterozygote deficiency). Thus, the data strongly suggest that these summer spawners represent a distinct population rather than admixture. Spawning time may be fine-tuned by the dosage of alleles affecting spawning time. The three populations from Gävle showed nearly identical allele frequencies at loci with strong genetic differentiation between Atlantic Ocean and Baltic Sea (*Figure 3C*), whereas they showed dramatic allele frequency differences at loci associated with spawning time (*Figure 4C*).

We used SNP-chip data to construct a haplotype tree based on highly differentiated SNPs in scaffold 190/1420. Two haplotype groups were strongly associated with spring- and autumn spawning (*Figure 4D*); haplotype trees for other top scaffolds are in *Figure 4—figure supplement 2*. The estimated average heterozygosity per polymorphic site across scaffold 1420 indicated a selective sweep among spring-spawning herring but not in autumn-spawning populations (*Figure 4E*). However, the nucleotide diversity did not show a significant difference between groups (spring: $0.24\% \pm 0.004\%$; autumn: $0.27\% \pm 0.003\%$). Thus, the number of variable sites are higher among spring-spawning herring, but the average heterozygosity per site is lower. One possible explanation for this observation is that a selective sweep happened at this locus in the past in spring-spawning herring, which was then followed by a population expansion allowing the accumulation of new mutations. This

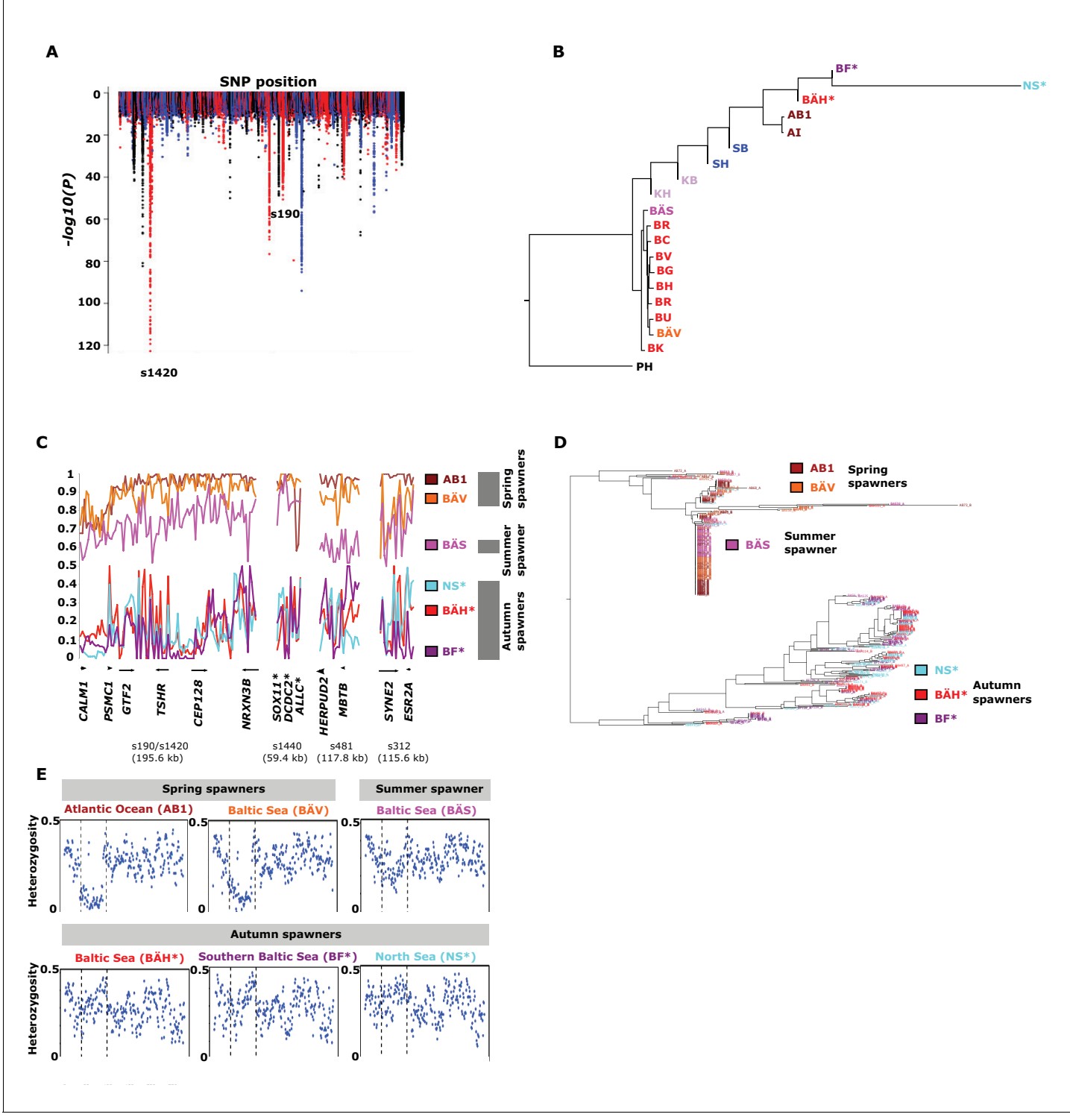

**Figure 4.** Genetic differentiation between spring- and autumn-spawning herring. (A) Manhattan plot of significance values testing for allele frequency differences. (B) Neighbor-joining phylogenetic tree based on all SNPs showing genetic differentiation in this comparison (p<10⁻¹⁰). (C) Comparisons of allele frequencies in four strongly differentiated regions. The major allele in the AB1 sample (Atlantic Ocean) was set as reference at each SNP. Scaffolds 190 and 1420 have been merged in this plot since it was obvious that they were overlapping. *The signal in scaffold s1440 is present ~27 kb upstream of *SOX11* and ~46 kb downstream of *DCDC2/ALLC*. (D) Neighbor-joining tree based on haplotypes formed by 70 differentiated SNPs from scaffold 190/1420; same populations as in *Figure 4C*. (E) Plot of average heterozygosity, per SNP in 5 kb windows, across scaffold 1420 indicating a selective sweep among spring-spawners in the region marked with vertical hatched lines. Autumn-spawning populations are marked by an asterisk.

*Figure 4 continued on next page*

*Figure 4 continued*

The following figure supplements are available for figure 4:

**Figure supplement 1.** Analysis of deviations from Hardy-Weinberg equilibrium using the $F_{IT}$ statistic in spring- (BÄV), summer- (BÄS) and autumn- (BÄH) spawners from the same locality (Gävle).

**Figure supplement 2.** Additional neighbor-joining trees for the contrast between spring- and autumn spawning herring.

interpretation is supported by strong negative Tajima's D-values in this region among spring-spawning Atlantic and Baltic herring (*Figure 1—figure supplement 1E*).

Genetic differences in spawning time are expected to involve photoperiodic regulation of reproduction. Interestingly, our strongest signals (p<1x10^{-120}) in this contrast is located within and up to 25 kb upstream of *TSHR* encoding thyroid-stimulating hormone receptor, which has a central role in this pathway in birds and mammals (*Nakao et al., 2008*; *Ono et al., 2008*; *Hanon et al., 2008*). Further, a second gene in the same scaffold (190/1420), *calmodulin* has a role in initiating reproduction following secretion of gonadotropin-releasing hormone (GnRH) (*Melamed et al., 2012*) downstream of TSHR signalling in photoperiodic regulation of reproduction. *SOX11*, one of the genes in the associated region in scaffold 1440 (*Figure 4C*), encodes a transcription factor that controls GnRH expression in GnRH-secreting neurons (*Kim et al., 2011*). Finally, *ESR2a*, in scaffold 312, encodes estrogen receptor beta that has a well established function in reproductive biology (*Bondesson et al., 2015*). Interestingly, a previous experimental study in sticklebacks also indicate that estrogen receptor signaling is involved in photoperiodic regulation of reproduction since treatment with aromatase inhibitors, which leads to an inhibition of the conversion of androgens to estrogens, altered photoperiodic regulation of male sexual maturation (*Bornestaf et al., 1997*). Also, the expression of *ESR2* but not *ESR1* is regulated by circadian factors in mice (*Cai et al., 2008*), consistent with our data suggesting that estrogen receptor beta (encoded by *ESR2*) is more important than estrogen receptor alpha (encoded by *ESR1*) for photoperiodic regulation of reproduction.

## Adaptive haplotype blocks are maintained by selection

A common feature for the signatures of selection for adaptation to low salinity and for seasonal reproduction in herring is the presence of haplotype blocks (10–200 kb in size) showing strong differentiation (*Figures 3C*, *4C*), despite the rapid decay of linkage disequilibrium at selectively neutral sites (*Figure 1—figure supplement 1A*). A possible explanation for the pattern is the presence of inversions suppressing recombination as previously shown in three-spined stickleback (*Jones et al., 2012*). We constructed 3.3 kb Nextera mate pair libraries for two Atlantic and two Baltic herring individuals to scan for inversions with a particular focus on regions under selection. However, few convincing inversion candidates were detected and none coincided with the regions highlighted in *Figures 3C*, *4C*. Thus, inversions do not appear to be an important explanation for the presence of haplotype blocks.

Having excluded inversions as a major explanation for the long haplotype blocks, two other possible explanations were considered. Haplotype blocks may occur as a consequence of recent fast selective sweeps that leads to hitchhiking of neutral polymorphism in close genetic linkage with causal variants (*Maynard-Smith and Haigh, 1974*; *Charlesworth et al., 1997*). Alternatively, haplotype blocks involving multiple causal mutations may be maintained by natural selection. These two models give entirely different predictions as regards nucleotide diversity in the differentiated regions of the genome. The hitchhiking model predicts reduced levels of genetic diversity in the differentiated region whereas the haplotype evolution model implies that nucleotide diversity in the differentiated regions, even within populations, may be as high or even higher than in neutral regions because the haplotypes are expected to have been maintained during an evolutionary process. We decided to test this by comparing nucleotide diversity for the 30 most differentiated regions in the contrast Atlantic vs. Baltic within and between one population of Atlantic herring (Bergen) and one population of Baltic herring (Kalix). The nucleotide diversity turned out to be significantly higher in the differentiated regions than in random regions of the genome both within and between populations (*Figure 5A*). The same conclusion emerged from the analysis of the 30 most differentiated regions between autumn- and spring-spawning herring using the samples collected at the same

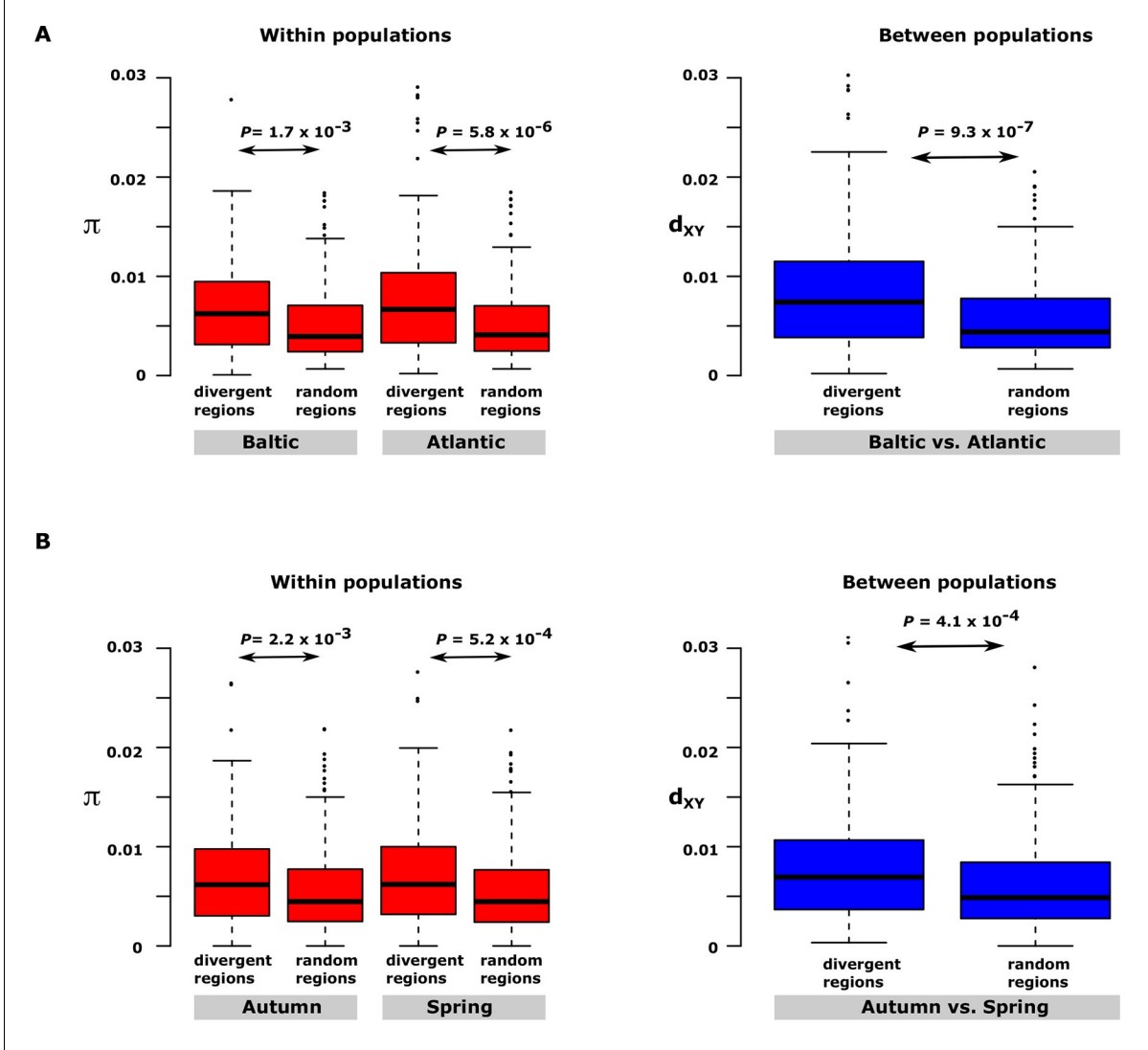

**Figure 5.** Nucleotide diversity within and between samples with different ecological adaptations as regards (**A**) salinity and (**B**) spawning time. For each contrast 30 strongly differentiated regions of the genome and 30 control regions showing no significant differentiation were used. The nucleotide diversity within and between populations for the control regions was estimated around 0.3% consistent with the genome average whereas diversity in differentiated regions was significantly higher. BK=Baltic herring, Kalix; AB=Atlantic herring, Bergen; BÄH=autumn-spawning Baltic herring from Gävle; BÄV, spring-spawning Baltic herring from Gävle; see *Table 1*. The data are presented as box plots; the central rectangle spans the first to third quartiles of the distribution, and the 'whiskers' above and below the box show the maximum and minimum estimates. The line inside the rectangle shows the median.

locality (Gävle) in May and September (*Figure 5B*). Thus, we conclude that our data on genetic differentiation in herring is consistent with the evolution of haplotype blocks harbouring multiple causal variants. The model also implies that the presence of multiple alleles containing different combinations of causal variants is expected.

## Genomic distribution of causal variants

Genome-wide analysis combined with strong signatures of selection enabled us to explore the genomic distribution of sequence polymorphisms underlying ecological adaptation. We carried out an enrichment analysis as previously used to identify categories of SNPs showing differentiation between domestic and wild rabbits (*Carneiro et al., 2014*). We calculated the absolute allele frequency difference (dAF) for different categories of SNPs in the two contrasts Atlantic vs. Baltic and

spring- vs. autumn spawning herring and sorted these into bins (dAF 0–0.05, etc.) for different categories of SNPs. In both contrasts the great majority of SNPs (>90%) showed a dAF lower than 0.10 (*Figure 6*, *Supplementary file 3E*).

Non-synonymous substitutions showed the most striking enrichment in both contrasts and showed a steady increase above dAF=0.15 reaching a two-fold enrichment at dAF>0.50 (*Figure 6*, *Supplementary file 3E*). This enrichment must reflect natural selection acting on the protein sequence because synonymous substitutions did not show a similar strong enrichment at high dAF. All non-synonymous substitutions showing dAF>0.50 in any of the two contrasts are compiled in *Supplementary file 3F*. A striking feature of this list is the common occurrence of multiple high dAF SNPs in the same gene. The 74 non-synonymous changes with dAF>0.50 in the contrast Atlantic vs. Baltic occur in only 29 different genes and the corresponding figure for the contrast spring- vs. autumn-spawning is 21 non-synonymous changes in 9 genes. We excluded the possibility that the presence of multiple non-synonymous changes in many of the genes was explained by errors in gene models (non-coding sequences annotated as exons) by a comparative analysis with other teleosts. We identified the orthologous position for about two thirds of the positions listed in *Supplementary file 3F*, the great majority of these (58/62) were annotated as coding sequence also in other species (*Supplementary file 3F*).

SNPs located in the 5'untranslated and 3'untranslated regions (UTRs) showed a more consistent enrichment compared to synonymous changes implying that this enrichment is unlikely to be caused entirely by close linkage to coding sequences under selection. Thus, changes in UTRs have contributed to ecological adaptation in the herring, most likely due to their role in regulating mRNA stability and translation efficiency. In this analysis we combined 5'UTR and 3'UTR SNPs to avoid too small classes for the extremely high dAF. However, an analysis based on all SNPs showing a dAF > 0.1 in the Atlantic vs. Baltic contrast and all SNPs showing a dAF > 0.2 for the spring- vs. autumn-spawning contrast demonstrated that both 5'UTR and 3'UTR SNPs are overrepresented at high dAF and the trend is particularly strong for 5'UTR SNPs (*Supplementary file 3G*).

The importance of regulatory changes underlying ecological adaptation is evident from the highly significant enrichment of SNPs within 5 kb upstream and downstream of coding sequences (*Figure 6*, *Supplementary file 3E*). Further, the excess is particularly pronounced within 1 kb upstream of the coding sequence where the promoter is expected to be located (*Supplementary file 3H*). The enrichment is not as high as for non-synonymous changes but this does not mean that regulatory changes are less important than coding changes because a much higher proportion of SNPs within the 5 kb region flanking coding sequences are expected to be selectively neutral compared with those causing non-synonymous changes. Thus, it is possible that the enrichment of non-coding SNPs would be much higher if there was a better annotation of the functional significance of non-coding sequences in Atlantic herring.

Intergenic and intronic SNPs were in general underrepresented among SNPs showing high dAF (*Figure 6*). For the most differentiated SNPs (dAF > 0.50) the intergenic SNPs showed a marked underrepresentation in the Atlantic – Baltic contrast (M=-0.64; p=5.1 x $10^{-25}$; *Supplementary file 3E*) while intronic SNPs were most underrepresented in the spring- vs. autumn-spawning contrast (M=-0.55; p=6.7 x $10^{-7}$; *Supplementary file 3E*).

We also explored the possibility that loss of function-mutations have contributed to ecological adaptation. We identified a total of 469 nonsense mutations but expect that many of these will be false predictions due to errors in the gene model. Eight predicted nonsense mutations had a dAF higher than 0.20 in one of the contrasts and were further examined. Seven of these were unlikely to be correct annotations since the positions were not annotated as coding in zebrafish, and the remaining one had a dAF of 0.21 but was far from statistical significance. Thus, we conclude that gene inactivation is not a common mechanism for ecological adaptation.

## Discussion

We have generated an Atlantic herring genome assembly and used this for a comprehensive analysis of the genetic basis for ecological adaptation. Hundreds of independent loci underlying ecological adaptation were revealed by comparing spring- and autumn-spawners as well as populations adapted to marine and brackish waters. The data show that both coding and non-coding changes

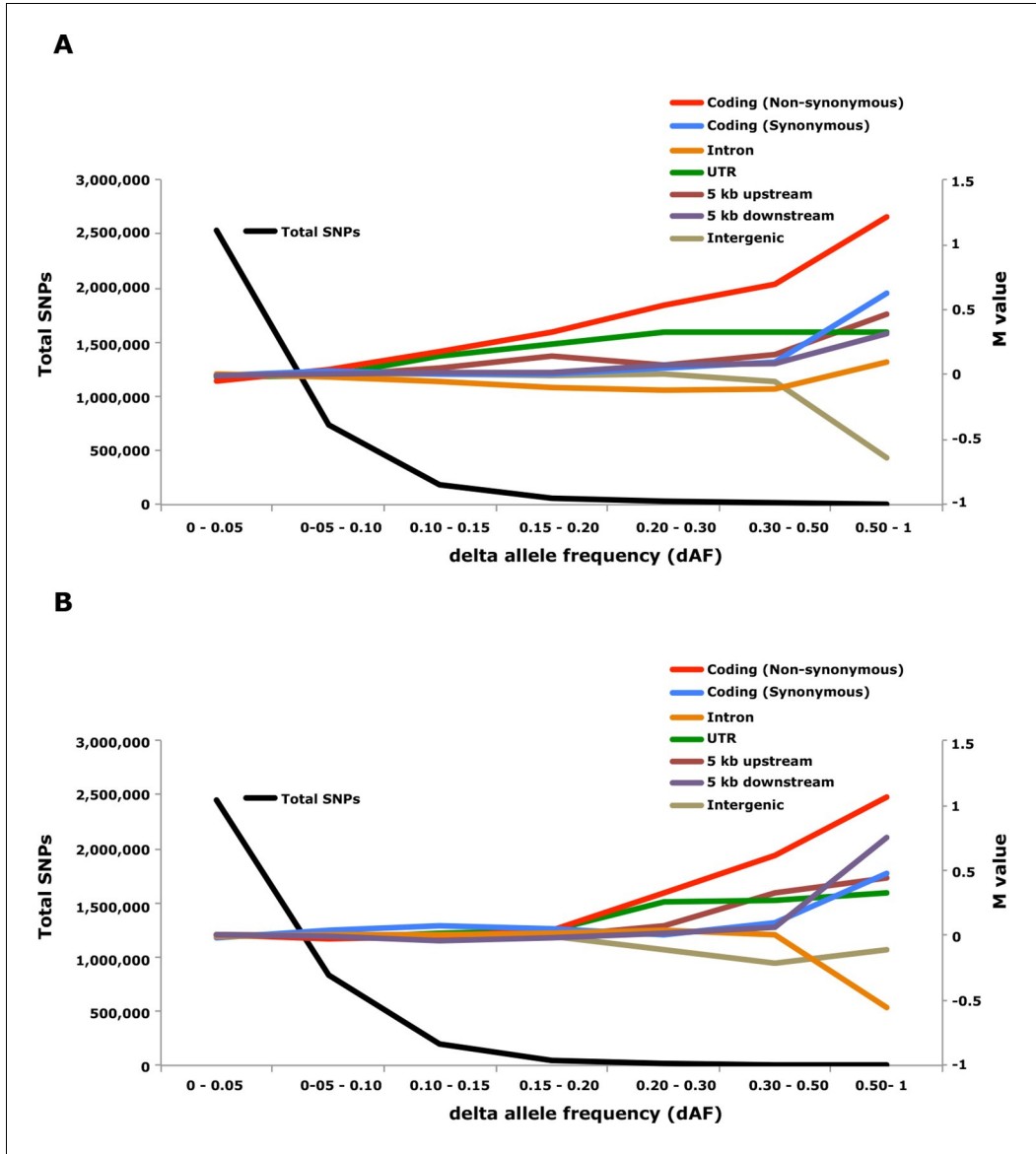

**Figure 6.** Analysis of delta allele frequency (dAF) for different categories of SNPs. (**A**) dAF calculated for the contrast marine vs. brackish water. (**B**) dAF calculated for the contrast spring- vs. autumn-spawning. The black line represents the total number of SNPs in each dAF bin and coloured lines represent M values of different SNP types. M values were calculated by comparing the frequency of SNPs in a given annotation category in a specific bin with the corresponding frequency across all bins.

contribute to ecological adaptation and we find that haplotype blocks spanning up to hundreds of kb show strong genetic differentiation.

The genetic architecture of multifactorial traits and disorders is an important topic in current biology. Genome-wide association studies (GWAS) in humans as well as in livestock have indicated that most multifactorial traits and disorders are controlled by large number of loci each explaining a tiny fraction of trait variation (*Wood et al., 2014*; *Meuwissen et al., 2013*). Thus, if ecological adaptation has a similar complex genetic background, in particular in a species with a large population size where each base in the genome is expected to mutate many times each generation, it may be difficult to reveal individual loci underlying adaptation. In contrast, this and our previous study (*Lamichhaney et al., 2012*) have revealed that genomic regions harbouring a small portion of all SNPs show strong genetic differentiation in the herring whereas the rest of the genome shows very

low levels of genetic differentiation. However, there are some important differences between the herring and human data. Firstly, human GWAS reveal loci that contribute to standing genetic variation and therefore includes deleterious alleles that have not yet been eliminated by purifying selection. Secondly, the phenotypic effects of the loci reported here in the herring may be small and the strong genetic differentiation may have accumulated over many generations. There is also plenty of room for natural selection to operate in a species with a large reproductive output like the herring. Thirdly, our study gives no insight in how much of the genetic variation in ecological adaptation these loci control since we do not have information on genotype-phenotype relationships for individual fish. We cannot exclude the possibility that there are additional loci with tiny differences in allele frequency between populations or loci with an extensive allelic heterogeneity that are not detected using our approach. The question how much of the genetic variation the loci reported in this study explains needs to be addressed in future experimental studies.

An important finding was the presence of large haplotype blocks (10–200 kb in size) showing strong genetic differentiation, standing in sharp contrast to the rapid decay of linkage disequilibrium at selectively neutral sites (*Figure 1—figure supplement 1A*). Although it is expected that the majority of sequence polymorphisms associated with these haplotype blocks are selectively neutral, the data presented here is consistent with a scenario where haplotype blocks evolve over time by the accumulation of multiple, consecutive mutations affecting one or more genes similar to the evolution of haplotypes carrying multiple causal mutations as has been documented in domestic animals (*Andersson, 2013*) as well as suggested for the evolution of the blunt beak *ALX1* haplotype in Darwin's finches (*Lamichhaney et al., 2015*). Under this scenario, the shift from one allelic state to another rarely happens through a single mutational event since the fitness of a haplotype depends on the combined effect of multiple sequence polymorphisms affecting function. Furthermore, it is expected that there will be selection for supressed recombination within these regions to avoid that favoured haplotype blocks break up. Our analysis showing that nucleotide diversity is higher within the differentiated regions than in the rest of the genome (*Figure 5*) strongly supports our hypothesis that the large haplotype blocks are maintained by selection rather than being the consequence of genetic hitchhiking (*Maynard-Smith and Haigh, 1974*; *Charlesworth et al., 1997*). The common occurrence of multiple non-synonymous changes in genes showing strong genetic differentiation provides further support for the haplotype evolution model (*Supplementary file 3F*). The model proposed here is in line with the evolution of complex adaptive alleles in species with large current effective population sizes like modern *Drosophila melanogaster* populations (*Karasov et al., 2010*).

A long-standing question in evolutionary biology is the relative importance of genetic variation in regulatory and coding sequences. *King and Wilson (1975)* argued already 40 years ago that regulatory changes are more important than protein changes for phenotypic differences among primates. The large number of loci associated with ecological adaptation detected in the present study allowed us to explore their genomic distribution. There was a highly significant excess of non-synonymous changes as well as SNPs in UTRs and within 5 kb upstream and downstream of coding sequences among the loci showing strong genetic differentiation (*Figure 6*). Thus, both coding and non-coding changes contribute to ecological adaptation in the herring. The enrichment was clearly most pronounced for non-synonymous SNPs but it is likely that regulatory changes are in majority among the causal variants because there are more than 10 times as many non-coding as coding changes among the SNPs showing the strongest genetic differentiation (*Supplementary file 3F*). However, at present we cannot judge the relative importance of coding and non-coding changes, partially due to the strong linkage disequilibrium between coding and non-coding changes and partially because we have no data on the effect size of individual loci. We observed a highly significant excess of several categories of SNPs even for loci with only a 10–15% allele frequency difference between populations (*Supplementary file 3E*) suggesting that SNPs with such minor changes in allele frequencies contribute to ecological adaptation in the herring. Consistent with previous studies in domestic animals (*Carneiro et al., 2014*; *Rubin et al., 2010*), we did not find any indication that gene inactivation has contributed to adaptive evolution.

Timing of reproduction is of utmost importance for fitness in plants and animals and it is well documented that climate change affects reproductive success in both terrestrial (*Visser et al., 2015*) and aquatic organisms (*Edwards and Richardson, 2004*). We identified more than 100 independent loci showing strong genetic differentiation between spring- and autumn-spawners. Not all of these are expected to control reproduction since other life history parameters differ between populations.

However, several of the most strongly associated regions overlapped genes with a role in photoperiodic regulation of reproduction in birds and mammals, such as *thyroid-stimulating hormone receptor (TSHR), calmodulin* and *SOX11* (*Ono et al., 2008*; *Hanon et al., 2008*; *Nakao et al., 2008*; *Melamed et al., 2012*; *Kim et al., 2011*). Photoperiodic regulation in fish is poorly studied, but a recent study showed that the saccus vasculosus brain region is a sensor of changes in day length and suggested that changes in day length affect *TSHR* expression in this region in Masu salmon (*Nakane et al., 2013*). Interestingly, strong signatures of selection at *TSHR* in chicken (*Rubin et al., 2010*) and sheep (*Kijas et al., 2012*) may reflect selection against seasonal reproduction in domestic animals.

The population structure of Atlantic herring has been under debate for more than a century (*McQuinn, 1997*; *Iles and Sinclair, 1982*). The discussion has concerned the taxonomic status of stocks associated with different spawning and feeding locations, and whether populations are reproductively isolated. Our data are consistent with a metapopulation structure (*McQuinn, 1997*) in which subpopulations (stocks) are not reproductively isolated. Gene flow combined with large effective population sizes explains low genetic differentiation at selectively neutral loci. Despite this, natural selection is sufficiently strong to cause genetic differentiation at many loci underlying adaptation.

Many populations of marine fish, including the herring, have been severely affected by overfishing (*Worm et al., 2006*; *Dickey-Collas et al., 2010*). Our study shows how genomic technologies can be used in a cost-effective manner to make major leaps in characterization of population structure and genetic diversity. The study has important implications for sustainable fishery management of herring by providing a comprehensive list of genetic markers that can be used for stock assessments, including the first molecular tools to distinguish autumn- and spring-spawning herring. These can be used to complement the current use of otoliths (ear bones) microstructures. Moreover, the findings that spring- and autumn-spawners constitute distinct populations imply that fisheries management should aim to protect both populations separately, which is currently not the case in the Baltic Sea (*ICES, 2014*). Finally, the study also has implications for fish aquaculture due to the interest to alter seasonal reproduction and adaptation to different salinities.

## Materials and methods

### Genome assembly and annotation

#### Sample collection

A single Baltic herring (*Clupea harengus membras*) captured at Forsmark, east of Uppsala, Sweden on September 21, 2011 was used as the reference individual. Skeletal muscle was isolated, placed in 20% glycerol and stored in -80°C until DNA preparation was performed. DNA extraction was carried out with a standard salt precipitation method without vortexing to generate high molecular weight DNA.

#### Genome sequencing and assembly

Libraries of eight different insert sizes from the reference individual were sequenced on Illumina HiSeq2000 and Illumina MiSeq (chemistry v2) to a total depth of 127-fold coverage of quality-filtered data (*Supplementary file 1A*). Reads were filtered according to the following criteria: we eliminated (i) read pairs that contained more than 10% Ns in one of their reads; (ii) read pairs with more than 40% low quality bases (quality ASCII-64 $\leq$ 7); (iii) read pairs containing adapter sequence (with mismatches $\leq$ 3bp); (iv) for those libraries with insert size longer than the sum of both reads, if reads overlapped, we skipped reads with at least 10 bp overlap and mismatch in more than 10% of the bases overlapping; (v) finally, we avoided reads showing to be PCR duplicates. Genome assembly with SOAP*denovo* v2.04 release 238 (*Li et al., 2010*) resulted in a scaffolded assembly of 834 Mb (*Supplementary file 1B*). An additional round of gap closing with an in-house pipeline was performed, utilizing both the overlapping HiSeq library and small unscaffolded contigs. To avoid short spurious contigs, which might appear in the assembly process, all contigs smaller than 1 kb were omitted from the final assembly. Potentially redundant contigs (no mismatches and maximal two gaps) were identified by self-aligning the complete assembly by BLAT, resulting in the removal of an additional 28 contigs. After aligning all reads back to the assembly, regions with no coverage (i.e. assembly valleys) were masked with N's (3.6 Mb in total). In addition, 298,968 nucleotide positions

where all the aligned reads conflicted with the assembly were corrected. The final assembly had a total size of 808 Mbp (*Supplementary file 1B*), which is in reasonable agreement with flow cytometry genome size estimates in Pacific herring (*C. pallasii*); 0.77–0.98 picograms (*Hinegardner and Rosen, 1972*, *Ida et al., 1991*, *Ohno et al., 1969*) corresponding to an average of ~850 Mb (mean (pg) x 0.978 $\times$ 10$^9$ bp/pg) (*Doležel et al., 2003*).

## Synthetic long reads

We obtained data from Illumina's synthetic long-read sequencing service (formerly Moleculo) (*McCoy et al., 2014*). Illumina prepared five libraries as a service following a detailed published protocol (*Voskoboynik et al., 2013*). In brief, pieces of genomic DNA of approximately 8 kb per molecule were distributed into 384 wells per library, with multiple molecules per well. The DNA in each well was fragmented, and sequencing adapters and well-specific barcodes were added. After pooling and sequencing on one lane of a HiSeq instrument, reads within each well were assembled separately in order to reconstruct the original molecules in each well. The contigs resulting from this process are called *synthetic long reads* (SLRs). They were filtered to be at least 1500 bp in size. From the five libraries, we obtained 1.3 million SLRs at an average length of 3.6 kb per read (4.7 Gb total sequence), with 20% longer than 5 kb (*Figure 2—figure supplement 3*). Since SLRs are assembled consensus sequences, base qualities were high, with an average Phred-scaled quality value of 36.

Reads were mapped with BWA-MEM 0.7.10 (*Li and Durbin, 2009*), with default parameters. The program was chosen since it was designed to work with long reads, and it also allows local alignments (split alignments of reads to different parts of the reference). This allows, for example, chimeric alignments in which the beginning and end of a read are aligned to different contigs. Overall, the resulting BAM file contained 2,743,529 alignments, corresponding to 2.09 alignments per read on average. There exist alignments for 99.95% of all long reads. Since this 'mapping rate' includes also reads that map only partially (due to local alignment), it is more informative to consider the fraction of *mapped bases* instead. We consider a base of a long read to be mapped if an alignment exists that includes that base. Under this definition, 4.47 of 4.69 Gbp (95.6%) could be aligned to the reference. On this level, we therefore observe a good agreement between reference and long reads.

Under ideal conditions, each long read would align to the reference in a single alignment that extends from the first to the last base of the read. Not being quite as strict, we considered a read to be fully aligned if there exists a single alignment of the read to the reference that involves at least 95% of the bases in the read. We observed 822,802 (62.68%) fully aligned reads. If we require 99% of the bases to be aligned, 771,192 (58.75%) reads remain. A limitation of this approach of assessing quality is that both the reference and the synthetic long reads are assemblies. An agreement between reference and SLR could therefore mean that both assemblies contain the same error. However, synthetic long reads are assumed to be more reliable since the assembly problem, being restricted to only a fraction of the whole genome, is inherently less complex for SLRs. On the other hand, disagreement between reference and SLR may represent normal variation between alleles, not necessarily an assembly error. Despite their length, SLRs did not improve the assembly since we observed that SLRs typically failed to assemble at the same loci at which the genome assembly contained gaps, preventing these gaps from being spanned and closed. The coverage from SLRs mapped to the assembly was also uneven, with 36.7% of genomic bases not being covered at all (*Figure 2—figure supplement 4*).

SLRs allowed an improved read-based phasing (*Kuleshov et al., 2014*) of the 3,896,765 variants called by FreeBayes (version 0.9.8) in the reference individual (not including SLRs). Using GATK's ReadBackedPhasing (*McKenna et al., 2010*), 61.0% of those variants could be phased, yielding blocks that contain 14.7 variants on average. Providing the program also with the synthetic long reads improves this to 22.8 variants per block on average. On average, the block size improves from 1,254.32 bp to 2,476.07 bp. The largest phased block consists of 824 variants without SLRs (length=81,861 bp), but 1664 variants when SLRs are included (length=188,052 bp).

Indels in the reference individual were called separately with Illumina data and with SLR data using FreeBayes (version 0.9.8). The availability of long, high-quality reads makes it possible to call medium-sized indels. This type of indels are too long to be found using short reads, but too short to be found by techniques relying on mate-pair mapping distance so by using long reads we should improve our chances of finding them. Calling variants on the reference individual from only short

reads results in 537,186 indels of quality 100 or better. Only 580 of those have a length of 30 bp or more. Feeding the variant caller with the SLRs resulted in 8,372 additional indels of length 30 bp or more (*Supplementary file 1F*).

## Mappability calculation

We calculated per base mappability, a measure of base pair uniqueness in the reference sequence, with the GEM library (*Marco-Sola et al., 2012*). We translated the program's output to a bounded score per base pair along the genome. Thereafter, we binned these scores using 1 kb non-sliding windows in order to identify regions of the genome that are either highly unique or repetitive. By doing so, we could collate this information track and other annotations when searching for structural changes to better interpret the results.

## CEGMA

The completeness of the assembled genome was evaluated by analysing a set of 248 ultra-conserved eukaryotic genes using hidden Markov models (HMM) as implemented in CEGMA (v2.4) (*Parra et al., 2007*, *2009*). 84% of the core genes were scored as 'complete' in the assembly (>70% aligned), and only 2.5% were missing from the assembly (<30% aligned), which indicates that the gene space is well represented in the assembly (*Supplementary file 1C*).

## RNA sequencing, transcriptome assembly and annotation

Liver and kidney tissue were collected from the reference individual and stored in RNAlater. After extraction of RNA, using Qiagen RNeasy Fibrous Tissue Mini Kit, and poly-A selection, strand-specific dUTP libraries were produced. Fragments with an insert size of 200 bp were then sequenced by Illumina Hi-Seq instrument using 101 cycles per run producing ~200 million paired-end reads for each library (*Supplementary file 1A*).

We reconstructed the herring transcriptome combining RNAseq data for liver and kidney, from the reference herring, in addition to the previously published transcriptome from skeletal muscle from another Baltic herring individual (*Lamichhaney et al., 2012*).

Both the Swedish genome annotation platform and the BGI team then carried out genome annotation using a custom annotation pipeline. This process utilised various programs detailed in *Figure 2—figure supplement 1*. A combination of data from evidence sources (protein homology, transcripts, repeats) and *ab initio* predictions were used to discover 23,336 coding gene models (*Supplementary file 1D*).

## Genome annotation

A custom annotation pipeline using the Maker package (version 2.31.6) (*Cantarel et al., 2008*) was applied to combine evidence data (protein homology, transcripts, repeats) and *ab initio* predictions into gene annotations (*Figure 2—figure supplement 1*). First, using TopHat2 (v2.0.9) (*Kim et al., 2013*) and cufflinks (v2.2.1) (*Trapnell et al., 2010*), we reconstructed individual genome-guided RNA-seq assemblies of the reference herring transcriptome from liver and kidney tissues, and the previously published transcriptome from skeletal muscle from another Baltic herring individual (*Lamichhaney et al., 2012*). In order to obtain a high-confidence set of coding transcripts, we set the minimum isoform fraction (-F) to 0.25 in cufflinks (this allows isoforms to be reported if they accounted for 25% of the expression in a given sample) and the pre-mRNA fraction (-j) to 0.6 (suppressing reads that are spanned by splice junctions and are expressed at 60% or lower when compared to the splice junction). While this approach may loose some data, we found it to yield satisfactory results with regards to noise levels and spurious transcription, judged by coding potential and structure (31,374 transcripts built from muscle, 50,404 from kidney and 41,285 from liver). In a complementary approach, we computed a *de novo* assembly of normalized, merged samples of liver and kidney, using the Trinity package (*Grabherr et al., 2011*) with default settings (248,721 transcripts).

We also collected further evidences at the protein level by querying the Uniprot database (*Magrane and Consortium, 2011*) for sequences belonging to the *Teleostei* taxonomic group (n=46,288 proteins). Proteins gathered had to be supported at either the proteomic or transcriptomic level and could not be fragmented. Additionally, we downloaded the Uniprot-Swissprot

reference data set (on 2014-05-15) (n=545,388 proteins) for a wider taxonomic coverage with high-confidence proteins. The two protein data sets yielded a total of 587,735 non-redundant sequences that provided guidance to the putative structure and CDS phases of annotated loci.

To increase the accuracy of the annotation and annotate repeats, we used the existing reference repeat library included in the RepeatMasker (v4.0.3) (*Smit et al., 2015*) enhanced by novel repeats detected with the RepeatModeler package (v1.0.8) (*Smit and Hubley, 2010*). Candidate repeat sequences identified were vetted against a set of putative transposon sequences contained within our protein data set (referred as the 587,735 non-redundant set above) to exclude any nucleotide motif stemming from low-complexity coding sequences. After augmenting this repeat library, we utilized RepeatMasker and RepeatRunner (*Stanke et al., 2008*) to assign repeat sequences to genomic loci. RepeatRunner is a program that integrates RepeatMasker with BLASTX allowing the analysis of highly divergent partial or complete repeats as well as protein coding parts within retroelements and retroviruses undetected by RepeatMasker. Overall, 30.9% of the assembly contains various families of repeats (*Supplementary file 1E*).

We generated different gene builds using the Maker pipeline (version 2.31.6) (*Holt and Yandell, 2011*). We first constructed a so-called evidence build, which was computed directly from the sequence data we had compiled (transcripts and proteins) without using any *ab initio* predictions. This yielded a set of 19,762 gene models and 24,551 mRNAs, subsequently referred to as candidate 4 (rc4). However, evidence-based annotation alone is limited by the available sequence data, potentially returning fragmented structures or missing loci entirely. To prevent this happening, we next performed an *ab initio* aided re-annotation of the initial gene build. Information from the evidence build was used to curate 1100 non-redundant, high-confidence transcript models (i.e. full length as judged by synteny to genes in zebrafish as well as transcriptome and protein alignments) and to train the augustus gene predictor (version 2.7) (*Stanke et al., 2008*).

With the aid of the *ab initio* model, we performed a second pass (re-annotation) and replaced any existing loci where a longer putative CDS could be predicted by the gene finder or filled in gene predictions where sufficient evidence was lacking for the construction of evidence models. This improved annotation was called rc5 and yielded 22,380 genes (and 26,259 mRNAs), adding 2767 novel loci compared to the previous rc4 build. In contrast, 357 loci had been omitted from the re-annotation but were present in the rc4 so we added them back to the rc5, upgrading the gene count to 22,737 (and 26,712 mRNAs). In order to verify the added value of this two-step build strategy, we utilized a quality check referred as Annotation Edit Distance (AED) comparing mRNAs from rc4 and rc5. This metric means to quantify the congruency between a gene annotation and its supporting evidence. The closer an AED score is to 0, the 'better' the annotation is. The comparison of the density distribution of AED scores between gene builds at this step (*Figure 2—figure supplement 2*) shows that the complementary usage of the *ab initio* predictor achieves an overall improved congruency between models and supporting evidence.

Finally, as an additional polishing step, we used the PASA package (*Haas et al., 2003*) in combination with de-novo assembled transcripts from the liver and kidney tissue samples with the aim to further refine individual gene models and check all rc5-derived transcript models for proper coding potential (*Haas et al., 2003*). This final gene set release (rc6) yielded a total count of 23,336 genes. Statistical evaluation of the final rc6 set was performed using the GAG package (*Hall et al., 2014*). Based on this data, around 8% of the genome retains coding potential (*Supplementary file 1D*). Apart from the protein coding genes, we also discovered a total number of 993 tRNAs with a total gene length of 73,203 bp using tRNAscan (v1.3.1) (*Lowe and Eddy, 1997*).

With the initial gene build completed, we proceeded to infer putative functions for all coding mRNAs. To this end, we first predicted functional domains using InterProscan (v5.7–48) (*Jones et al., 2014*) to retrieve Interpro (*Hunter et al., 2012*), PFAM (*Finn et al., 2014*) and GO (*Ashburner et al., 2000*) annotations. In order to assign protein and gene names to this dataset, we performed a BLASTp (version 2.2.28+) search with each of the predicted protein sequences against the Uniprot/Swissprot reference data set (downloaded on 2014-05-15) with default e-value parameters ($1 \times 10^{-5}$). Outputs from both analyses were parsed using the Annie annotation tool (*Tate et al., 2014*) to extract and reconcile relevant metadata into predictions. Only 4,838 transcripts (3,375 genes) remained with no functional information available.

The standard annotation pipeline combining evidence based and *ab initio* predictions applied to our present assembly predicted a total of 23,336 gene models in the herring genome

(*Supplementary file 1D*). We performed clustering of orthologous genes among 15 species (*H. sapiens, D. rerio, L. chalumnae, G. morhua, T. rubripes, T. nigroviridis, O. latipes, C. harengus, G. aculeatus, P. marinus, O. niloticus, L. oculatus, X. maculatus, A. mexicanus, P. formosa*), downloaded from Ensembl 78, with OrthoMCL (version 2.0.9) and used granularity of 1.5 as recommended for the mcl algorithm (*Li et al., 2003*). We first filtered the proteomes to keep only the longest isoform per gene. Then we filtered by length, keeping only those with more than 50 amino acid residues. None of the herring proteins were removed but 208 proteins from the three proteomes of the 4way-fish comparison were removed. We found that the herring assembly contained 14,107 orthologous gene families, 9,634 of which were common to four fish genomes (*C. harengus, D. rerio, L. chalumnae, G. morhua*), and 573 of these gene families were specific only to the herring genome (*Figure 2B*). The difference between both figures is likely to be a reflection of the different annotation statuses for current fish genomes, some of them more fragmented and poorly annotated, possibly yielding some spurious clustering in the 15way comparison.

## Endogenous retroviruses (ERVs)

Analyses of the herring genome using RetroTector (*Sperber et al., 2007*) identified 150 endogenous retroviruses (ERVs) in scaffolds or contigs larger than 12 kb constituting about 0.13% of the genomic sequence (*Supplementary file 2*), none of which presented open reading frames in all *gag, pol* and *env* genes. The number of identified ERVs is somewhat lower than in most vertebrate hosts but comparable to other fish genomes (*Hayward et al., 2015*). Epsilon retroviruses such as the Walleye dermal sarcoma virus (WDSV) are typical findings in fish genomes and 8 epsilon-like ERVs could be determined by phylogenetic analysis (*Figure 2—figure supplement 5*). Additionally, three ERVs group together with the Snakehead fish retrovirus (SnRV) and 51 ERVs form a large basal clade in the phylogenetic tree without known reference sequences, possibly a transition between known retroviral sequences and gypsy retrotransposons represented by Cer1 in the root of the tree.

## Genome resequencing and data analyses

### Sampling

Tissue samples from 47–100 fish per population were collected from different localities in the Baltic Sea, Skagerrak, Kattegat, the Atlantic Ocean and the Pacific Ocean (*Table 2*). Genomic DNA was isolated by standard procedures and DNA from all individuals per sampling location was pooled in equimolar concentrations. We also used eight DNA samples of Baltic herring collected close to Gävle, Sweden and eight Atlantic herring collected close to Bergen, Norway for individual sequencing.

### Resequencing, alignment and SNP calling

Sequencing libraries (average fragment size about 400 bp) were constructed for each population pool and the 16 individuals, and 2x100 bp paired-end reads were generated using Illumina HiSeq2000 sequencers. The amount of sequence per pool was targeted to ~30x coverage. These sequences in addition to the data from eight herring populations from our previous study (*Lamichhaney et al., 2012*) (*Table 2*) were aligned to the herring reference genome using BWA-MEM v0.7.1 (*Li and Durbin, 2009*). SNP calling was done using a standard GATK pipeline (*McKenna et al., 2010*). The quality filtering of the raw variant calls was done using GATK using the following cut-offs, QD < 2.0, MQ < 40.0, FS > 60.0, MQRankSum <-12.5, ReadPosRankSum < -8.0 and DP < 100. In addition, only SNPs with an average sequence coverage of 30-50x in each pool were retained for downstream analysis to avoid regions of the genome that are difficult to sequence using current technology and regions oversampled due to for instance duplications. Similar filtering criteria were used for individual sequences using GATK (QUAL < 100, MQ < 50.0, MQRankSum < -4.0, ReadPosRankSum < -2.0, QD < 2.0, HaplotypeScore > 10.0, FS > 60.0, DP < 12, DP > 720.0).

### Population genetics and demographic history

The filtered SNP dataset was used to estimate genetic diversity within and between populations using Plink (*Purcell et al., 2007*) and to generate neighbor-joining trees using Phylip (*Felsenstein, 1989*). The split between Atlantic and Pacific herring was dated based on sequence divergence for mtDNA *cytochrome B* sequence using the molecular clock calibration for this sequence

from fish (*Burridge et al., 2008*). Average nucleotide diversity was calculated using the 16 individual sequences, by counting the number of heterozygous sites in each individual and dividing by the total length of the used scaffolds. In order to avoid edge effects, only scaffolds longer than one Mb were included in the calculation. Decay of linkage disequilibrium, measured as correlation between genotypes, and Tajima's D were calculated using VCFtools (*Danecek et al., 2011*).

In order to compare the observed allele frequency distribution with the expected one at genetic equilibrium we performed a simulation study. We first used the coalescence simulation software 'ms' (*Ewing and Hermisson, 2010*) to generate 100,000 independent segregating loci in a large population (*n*=1000) under either a constant population size model (command: "ms 1000 100000 -s 1"), or a model with an expansion event in the recent past (command. "ms 1000 100000 -s 1 -eN. 1. 35"). Thereafter we randomly sampled 32 chromosomes in order to get a distribution that was comparable to our empirical sample, which contains data from 16 individuals. The sampling procedure was repeated 10 times, allowing for estimation of the variation in each allele frequency bin. However, due to the large number of sampled loci, this was close to zero. The demographic history of the Atlantic herring was explored using the diCal software (*Sheehan et al., 2013*). This software implements an extension of the Pairwise Sequentially Markovian Coalescent (PSMC) method (*Li and Durbin, 2009*). The extension makes it possible to jointly analyse several individuals, which increases the total number of coalescence events and thus improves resolution, particularly for recent time periods. In absence of mutation rate rates from Atlantic herring or any closely related species, we used a mutation transition matrix based on human-chimp data, while the point mutation rate and the recombination rate per base were assumed to be $1.25 \times 10^{-8}$, in accordance with *Sheehan et al. (2013)*. For the discrete intervals at which effective population size was estimated (the input '*p*' parameter in diCal), we used the string "3+2+2+2+2+2+2+2+2+2+3", for a total of 11 independent time periods.

We analysed the Baltic and Atlantic populations separately, using phased sequence data from eight individuals (i.e. 16 unique chromosomes) from each population. Due to memory usage constraints, we performed the analysis for a limited number of genomic regions, each containing approximately $10^5$ SNPs, arbitrarily chosen from parts of the genome that did not show strong signs of selection. A representative population history was then reconstructed by averaging across the analysed regions.

## Screening for signatures of selection

We classified populations based on their adaptations to different environmental variables, marine (Atlantic Ocean) vs. brackish (Baltic Sea) waters, and different spawning seasons (spring vs. autumn). We then performed contingency $\chi^2$ tests using the allele frequency estimates at each locus to identify genomic regions showing significant allele frequency differences between populations sorted according to these contrasts. In order to control for the inflation in *P* values at positions with high coverage, we normalised the reference and variant allele read counts at these positions using genome-wide expected coverage. The Bonferroni correction threshold was $p=8.2\times10^{-9}$ and $p<1\times 10^{-10}$ was chosen as the stringent significance threshold. We also estimated pooled heterozygosity as previously described (*Rubin et al., 2010*) in 5 kb genomic window across the whole genome in each population to identify genomic regions with reduced heterozygosity that may have been targets of selection. All significant SNPs associated with spawning or adaptation to variation in salinity ($p<10^{-20}$) were clustered as one independent genomic region under selection using the following steps. 1) We rescaled the coordinates by subtracting gaps from SNP positions along each scaffold. 2) We combined strongly correlated SNPs using the Comb-p software (*Pedersen et al., 2012*) and requested that independent regions should be separated by a distance of at least 20 kb with no SNPs reaching the significance threshold ($p<10^{-20}$).

Bayenv2 (*Günther and Coop, 2013*) was used to scan the genome for correlation between salinity and population differentiation. This Bayesian method tests for correlation between allele frequency patterns and environmental factors; we used the salinity at each sampling locality as given in *Table 2*. First, we generated a variance-covariance matrix among all populations (except Pacific) with 100,000 Markov Chain Monte Carlo (MCMC) iterations based on a subset of randomly chosen SNPs (2500). Then, we calculated the correlation between allele frequencies and salinity across all 19 populations for significant SNPs (46,045 SNPs) detected using the $\chi^2$ test by running 100,000

MCMC iterations. Bayenv2 reports two statistics: i) Bayes' factor (BF) and ii) Pearson correlation ($r$). We defined the significance threshold of these two statistics based on their distribution; $\log10 (BF) \geq 4$ and $|r| \geq 0.8$.

In order to identify the loci showing the most consistent allele frequency differences between autumn and spring spawning herring we used a generalized linear mixed model analysis (GLMER, implemented in the R package 'lme4') (*Bates et al., 2014*), which took into account the variability between populations within groups. The following R-code was used:

glmer(y ~ Group+(1|populations), family=binomial(), nAGQ = 10, data=input_file), where, Group = autumn or spring spawning population, Population = specific population, Family = error distribution used in the model, nAGQ = the number of points per axis for evaluating the adaptive Gauss-Hermite approximation to the log-likelihood, Default = 1, that corresponds to the Laplace approximation. As the differences in allele frequencies between the populations within the groups were small in our dataset, we did not use the Laplace approximation, but used a greater nAGQ (i.e. 10) for more thorough likelihood maximization procedure. The Bonferroni correction threshold was $p=4.9 \times 10^{-6}$ and cut-off of $p < 1 \times 10^{-10}$ was chosen as the stringent significance threshold.

## Genomic distribution of genetic variants

SnpEff (v.3.4) (*Cingolani et al., 2012*) was used to annotate the genomic distribution of variants and classify them into different categories (non-synonymous, synonymous, UTR, 5 kb upstream, 5 kb downstream, intronic and intergenic). For both (1) Atlantic vs. Baltic and (2) spring- vs. autumn-spawning contrasts, we calculated the absolute allele frequency difference (dAF) and sorted them into bins (dAF -0.05, etc.) for each of these categories of SNPs. For unbiased estimation of dAF, SNPs with missing calls in at least one population were removed. The SnpEff prediction from less confident annotations (for instance, missing start and stop codons in the transcript) were excluded from the analysis. The expected number of SNPs for each category in each bin was calculated as p(category) X n(bin), where p(category) is the proportion of a specific SNP category in the entire genome and n(bin) is the total number of SNPs in a given bin. Log2 fold change of the observed SNP count for each category in each bin was compared against the expected SNP count (M-value) and statistical significance of the deviations from the expected values was tested with a standard $\chi^2$ tests.

## Detection of structural changes

We extracted depth of coverage for all populations using GATK:DepthOfCoverage (*McKenna et al., 2010*), after filtering out reads with mapping quality below 20. We compared populations using 1 kb non-overlapping windows where all pools were normalized against the AB1 sample that showed highest average depth. In short, we created a correction factor per population and applied it on the depth of coverage value for each window. For all the contrasts, we performed an analysis of variance (ANOVA) as described (*Carneiro et al., 2014*).

We compared populations for two contrasts: 1) Atlantic vs. Baltic and 2) spring vs. autumn spawning. For the Atlantic-Baltic contrast we scanned 878,278 windows of which 79,809 windows were excluded due to low depth across all populations. 3,780 windows showed significant difference with a $p < 0.001$. Stringent filtering based on $p < 0.001$ and an |M-value|>0.6 resulted in the detection of 707 loci of which 491 had mappability above 0.5. The size of the structural changes ranged from 1–26 kb (*Supplementary file 3B*). In the second contrast, spawning, we compared populations for spawning time where we analysed 799,346 windows after filtering low depth regions. With the same criteria applied in salinity contrast, we identified 69 loci ranging in size from 1–19 kb (*Supplementary file 3D*).

We searched for inversions using mate-pair (2x100 bp) Nextera libraries (Illumina) with an average insert size of 3–4 kb generated from four individuals: two Atlantic (one female and male) and two Baltic (one female and male) sequenced to 3X coverage using Illumina HiSeq2000, After trimming and filtering low quality reads, we aligned the reads on herring genome by BWA-MEM (*Li and Durbin, 2009*). We used DELLY (*Rausch et al., 2012*) and BreakDancer (*Chen et al., 2009*) with default parameters, except that mapping quality was set to 10. We used bitwise flag in BAM files to extract deviant reads indicative of inversions overlapping sweep regions.

## Genotyping of individual fish using high density SNP array

We designed an Affymetrix custom genotyping array with 72,560 SNPs, tiling the best strand for each SNP. Due to the fact that A/T and G/C SNPs require twice the features on the array that other markers require, the array layout covered 82,569 probe pair sets. These covered the majority of SNPs showing significant genetic differentiation together with the best 2000 monomorphic nucleotide sites to validate the individual plate/sample performance (in the dish quality metric, DQC). We submitted 36-mer nucleotide sequences around target SNPs to the manufacturer. SNPs flanked by other sequence polymorphisms were avoided as well as regions containing repetitive or low mappability sequences, Finally, we deprioritized A/T and C/G SNPs, as they take twice as much room on the array. Array experiments were performed by the Array and Analysis Facility, SciLifeLab (Uppsala, Sweden) according to standard protocol (Affymetrix Axiom 2.0 Assay Manual Workflow User Guide, P/N 702990 Rev3, Affymetrix).

Genotype calling was performed using the Affymetrix Power Tools (APT, version 1.15.2) followed by SNP-Polisher (version 1.4.0), according to the Best Practice Analysis Workflow described in Axiom Genotyping Solution Data Analysis Guide (P/N 702961 Rev. 2, Affymetrix). The DQC threshold was set to 0.95 according to the manufacturer´s instructions, based on previous data from herring arrays. Only the samples passing a Sample Call Rate (CR)>97% were retained after first genotyping round and were subjected to genotyping step 2. Twelve samples were excluded when using both DQC (n=2) and CR (n=10) filtering. At this point, all plates run were considered to be high quality plates. We utilized SNPolisher, an R package, for the purpose of genotyping the remaining samples. SNPolisher generates the different groups of clusters, AA, AB, BB and off target variant (OTV), with corresponding cluster plots for each SNP and evaluate quality. Thereby, this post-process analysis identifies the best probeset and assign to each one of six categories (PolyHighRes, MonoHighRes, NoMinorHom, CallRateBelowThreshold, OTVs, Other). It also calculates QC metrics for each SNP, classifies SNPs into categories, changes dubious SNP calls, and tests for intensity shifts between batches. We took only converted polymorphic SNPs (PolyHighRes, n= 41,575) into account for further analyses. Genotypes at these positions are referred to as high quality genotypes.

This SNP array was used to genotype 360 individual samples, 30 random fish from 12 of the populations included in pooled sequencing. A total of 348 samples retained enough quality to be used for downstream analysis.

## Acknowledgements

We are grateful to Greg Barsh, Nick Barton, Örjan Carlborg, Miguel Carneiro, Peter and Rosemary Grant, Magnus Nordborg, Lars Rönnegård, Matthew Webster and an anonymous reviewer for valuable comments on the manuscript, and to Sam Barsh and David Tengbjörk for expert technical assistance. The work was funded by the ERC project BATESON, the Swedish Research Council and Formas. The BGI group was supported by the Special Project on the Integration of Industry, Education and Research of Guangdong Province (2013B090800017) and the Shenzhen Special Program on Future Industrial Development (JSGG20141020113728803). AMB, MM, DE and BN are supported by a grant from the Knut and Alice Wallenberg Foundation to the Wallenberg Advanced Bioinformatics Infrastructure. Sequencing was performed by the SNP&SEQ Technology Platform, supported by Uppsala University and Hospital, SciLifeLab and Swedish Research Council (80576801 and 70374401). SNP typing was performed by the Array and Analysis Facility, SciLifeLab, Uppsala, Sweden. Computer resources were provided by UPPMAX, Uppsala University.

## Additional information

### Funding

| Funder | Grant reference number | Author |
| --- | --- | --- |
| European Research Council | Bateson | Leif Andersson |
| Swedish Research Council Formas | Research grant | Leif Andersson |

Knut och Alice Wallenbergs Research grant Leif Andersson
Stiftelse

The funders had no role in study design, data collection and interpretation, or the decision to submit the work for publication.

## Author contributions

AMB, Contributed to the assembly and genome annotation, Performed bioinformatic analysis of population data, Wrote the paper; SL, Performed bioinformatic analysis of population data, Wrote the paper; GF, HZ, JD, DE, MH, PJ, MM, BN, XLi, WC, XLia, CS, YF, KM, XZ, SM-YL, XX, Contributed to the assembly and genome annotation; NRa, MP, C-JR, Performed bioinformatic analysis of population data; CF, UG, Performed experimental work; MSA, Analysis and interpretation of data, Contributed unpublished essential data or reagents; MB, MC, AF, LL, NRy, Provided population samples; LA, Conceived the study, Led bioinformatic analysis of population data, Wrote the paper, Acquisition of data, Analysis and interpretation of data

## Author ORCIDs

Alvaro Martinez Barrio, http://orcid.org/0000-0001-5064-2093
Sangeet Lamichhaney, http://orcid.org/0000-0003-4826-0349
Nima Rafati, http://orcid.org/0000-0002-3687-9745
Marcel Martin, http://orcid.org/0000-0002-0680-200X
Leif Andersson, http://orcid.org/0000-0002-4085-6968

# Additional files

### Supplementary files

• Supplementary file 1. (A) Summary of sequencing data used to generate the herring genome assembly. (B) Statistics of the herring genome assembly developed using SOAP*denovo*. v1.0 to v1.4 refer to different versions of the assembly. Full description of the assembly process is provided within the Methods section. (C) Statistics for genome completeness based on 248 annotated core eukaryotic genes (CEGs) by CEGMA. (D) Statistics for the annotated protein-coding genes. (E) Statistics for annotated repeats. (F) Comparison of indels discovered using Illumina short reads or synthetic long reads (SLRs) in the reference herring genome.

• Supplementary file 2. Endogenous retroviruses detected with RetroTector.

• Supplementary file 3. (A) List of loci showing strong genetic differentiation between Atlantic and Baltic herring ($p<10^{-10}$). (B) List of structural changes associated with genetic differentiation between Atlantic and Baltic herring. (C) List of loci showing strong genetic differentiation between spring- and autumn-spawning herring ($p<10^{-10}$). (D) List of structural changes associated with genetic differentiation between spring- and autumn-spawning herring. (E) Genomic distributions of different categories of SNPs in different delta allele frequency (dAF) bins. Only SNPs called in all populations that had confident annotations were used in this analysis. (F) Non-synonymous substitutions showing strong genetic differentiation, delta allele frequency (dAF) >0.5. (G) Analysis of delta allele frequencies (dAF) for SNPs in 5'UTR and 3'UTR. Only SNPs called in all populations that had confident annotations were used in this analysis. (H) Analysis of delta allele frequencies (dAF) for SNPs within the 5 kb region upstream of start of transcription.

### Major datasets

The following datasets were generated:

| Author(s) | Year | Dataset title | Dataset URL | Database, license, and accessibility information |
|---|---|---|---|---|
| Martinez Barrio A, Lamichhaney S, Fan G, Rafati N, Pettersson M, Zhang H, Dainat J, Ekman D, Höppner M, Jern P, Martin M, Nystedt B, Liu X, Chen W, Liang X, Shi C, Fu Y, Ma K, Zhan X, Feng C, Gustafson U, Rubin C, Sällman Almén M, Blass M, Casini M, Folkvord A, Laikre L, Ryman N, Ming-Yuen Lee S, Xu X, Andersson L | 2016 | Herring genome | http://www.ncbi.nlm.nih.gov/assembly/318681 | Publicly available at NCBI Assembly (accession no. GCA_000966335.1) |
| Martinez Barrio A, Lamichhaney S, Fan G, Rafati N, Pettersson M, Zhang H, Dainat J, Ekman D, Höppner M, Jern P, Martin M, Nystedt B, Liu X, Chen W, Liang X, Shi C, Fu Y, Ma K, Zhan X, Feng C, Gustafson U, Rubin C, Sällman Almén M, Blass M, Casini M, Folkvord A, Laikre L, Ryman N, Ming-Yuen Lee S, Xu X, Andersson L | 2016 | Population sequencing | http://www.ncbi.nlm.nih.gov/sra/?term=SRP056617 | Publicly available at the NCBI Sequence Read Archive (accession no. SRP056617) |
| Martinez Barrio A, Lamichhaney S, Fan G, Rafati N, Pettersson M, Zhang H, Dainat J, Ekman D, Höppner M, Jern P, Martin M, Nystedt B, Liu X, Chen W, Liang X, Shi C, Fu Y, Ma K, Zhan X, Feng C, Gustafson U, Rubin C, Sällman Almén M, Blass M, Casini M, Folkvord A, Laikre L, Ryman N, Ming-Yuen Lee S, Xu X, Andersson L | 2016 | Data from: The genetic basis for ecological adaptation of the Atlantic herring revealed by genome sequencing | http://dx.doi.org/10.5061/dryad.5r774 | Available at Dryad Digital Repository under a CC0 Public Domain Dedication |
| Martinez Barrio A, Lamichhaney S, Fan G, Rafati N, Pettersson M, Zhang H, Dainat J, Ekman D, Höppner M, Jern P, Martin M, Nystedt B, Liu X, Chen W, Liang X, Shi C, Fu Y, Ma K, Zhan X, Feng C, Gustafson U, Rubin C, Sällman Almén M, Blass M, Casini M, Folkvord A, Laikre L, Ryman N, Ming-Yuen Lee S, Xu X, Andersson L | 2016 | Population sequencing | http://www.ncbi.nlm.nih.gov/sra/?term=SRP017094 | Publicly available at the NCBI Sequence Read Archive (accession no. SRP017094) |

| | | | | |
|---|---|---|---|---|
| Martinez Barrio A, Lamichhaney S, Fan G, Rafati N, Pettersson M, Zhang H, Dainat J, Ekman D, Höppner M, Jern P, Martin M, Nystedt B, Liu X, Chen W, Liang X, Shi C, Fu Y, Ma K, Zhan X, Feng C, Gustafson U, Rubin C, Sällman Almén M, Blass M, Casini M, Folkvord A, Laikre L, Ryman N, Ming-Yuen Lee S, Xu X, Andersson L | 2016 | Population sequencing | http://www.ncbi.nlm.nih.gov/sra/?term=SRP017095 | Publicly available at the NCBI Sequence Read Archive (accession no. SRP017095) |
| Martinez Barrio A, Lamichhaney S, Fan G, Rafati N, Pettersson M, Zhang H, Dainat J, Ekman D, Höppner M, Jern P, Martin M, Nystedt B, Liu X, Chen W, Liang X, Shi C, Fu Y, Ma K, Zhan X, Feng C, Gustafson U, Rubin C, Sällman Almén M, Blass M, Casini M, Folkvord A, Laikre L, Ryman N, Ming-Yuen Lee S, Xu X, Andersson L | 2016 | Population sequencing | http://www.ncbi.nlm.nih.gov/sra/?term=SRP056617 | Publicly available at the NCBI Sequence Read Archive (accession no. SRP056617) |

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
