## [Decision Letter]

[Editors’ note: a previous version of this study was rejected after peer review, but the authors submitted for reconsideration. The first decision letter after peer review is shown below.]

Thank you for submitting your work entitled "The genetic basis for ecological adaptation revealed by genome sequencing of the Atlantic herring" for consideration by *eLife*. Your article has been reviewed by three peer reviewers, including Nick Barton and Magnus Nordborg, who is a member of our Board of Reviewing Editors. The evaluation was overseen by Diethard Tautz as the Senior Editor. Our decision has been reached after consultation between the reviewers. Based on these discussions and the individual reviews below, we regret to inform you that this manuscript cannot be considered further for publication in *eLife*. However, we would consider a completely reworked resubmission which addresses the concerns of the reviewers, and in particular goes much further in distinguishing between causal from non-causal alleles.

The consensus opinion of the reviewers is that this is a really interesting study in an excellent study system, but that the analysis is insufficient and several of the conclusions not supported. In particular, the reviewers were not convinced that you can actually identify the causal sites, and several conclusions rest on this. However, exploring the limits of what may be concluded is also of great interest, especially given the quality of the data.

In essence, the analysis does not do the data justice.

Reviewer #1:

This is a substantial piece of work addressing an important question using a beautiful and promising system. It is certainly the case that identifying loci that show evidence having been under selection is much easier in gigantic population with little population structure, as appears to be the case for herring.

Unfortunately, the paper suffers from several conceptual/logical flaws and I do not think the major conclusions are supported.

The confusion begins in the first sentence of the abstract, where it is stated that "Ecological adaptation is of major relevance…, but the underlying genetic factors are typically hard to study in natural populations due to confounding between population structure and signatures of selection".

It is indeed true that population structure makes it hard to identify signatures of selection, but who says that genes involved in ecological adaptation will exhibit such signatures? Generally, such signals arise as a consequence of strong selection on individual genes. For a quantitative trait, is not clear that there will be any such genes, and is also not clear that any such genes found will explain much of the variation. Furthermore, there is no phenotype in the present study, nor any heritability. Thus the comparison with human height (Discussion section) does not make sense. You have no idea how much of the variation for fitness your SNPs explain. Indeed it is formally possibly that all your SNPs are associated with defense against a parasite with a life-cycle that depends on salinity. I obviously don't believe this, but the point remains that in order to discuss the genetic architecture of adaptation, there needs to be genetics in the study. For example, a reciprocal transplant study that quantified the fitness effects of the identified loci.

Speaking of associations, I don't understand how the threshold for calling frequency differences significant was set (Figure 1—figure supplement 1). This is not explained ("used a QQ-plot" is not an explanation).

It is also stated that the presence of haplotype blocks was striking. Why? Presumably these are simply the consequence of selection? Are they longer than you would expect? Why? What is the evidence that they contain multiple causal sites as opposed to being the result of linkage drag?

Regardless of why they are there, the existence of these blocks mandate extreme caution when trying to decide which sites are causal. It is by now clear that local haplotype structure, especially when coupled with allelic heterogeneity causes large number of local spurious associations, and that, as a consequence, the most strongly associated SNP is NOT likely to be the causal one: it is likely to be an intermediate-frequency SNP that happens to tag the underlying variants well…

A consequence of this is that most of the conclusions about what kinds of SNPs do what (Subsection “Genomic distribution of causal variants”) are not supported. You find few significantly associated non-synonymous substitutions, but this could simply be because they are all rare, and can only be found by proxy.

It does make sense to compare different kinds of regions, as you do, but everything has to be controlled for allele frequency.

On a more population genetics level, I was struck by your low nucleotide diversity coupled with a rapid decay of linkage disequilibrium. Unless mutation- and recombination rates are very different from other organisms, they make no sense. My guess is that your nucleotide diversity is underestimated because you are detecting heterozygotes in single individuals, and that your rate of linkage disequilibrium decay is overestimated because of bad SNPs.

Third paragraph subsection “Genetic basis underlying timing of reproduction”: This something not right about this argument. Why would rare SNPs affect nucleotide diversity one way or another?

Subsection “Genomic distribution of causal variants”: "Some of these will be false positives due to close linkage to true positives but many will be true positives" I discuss this above, but it would be important to realise that, without assumptions about the meaning of "some" and "many", it is not possible to conclude any of what you conclude further down.

Same section: "Enrichment" compared to what?

Reviewer #2:

This paper uses whole-genome sequencing from 20 population pools, supported by SNP genotyping of ~360 individuals, to identify variants associated with differences between Baltic and Atlantic, and between autumn and spring spawning herring. This is an impressive set of data, and as the authors argue, the recent divergence, large population size and low differentiation makes herring exceptionally well suited to identifying genes responsible for adaptation. Thus, the study could in principle be a very nice contribution to *eLife*. However, the interpretation of the results seems quite naive: more detailed and rigorous argument is needed to be convincing.

SNP and structural variants associated with divergence are clustered into haplotype blocks that may span multiple genes, and up to 200kb. It seems to be assumed that all (or at least, many) of the variants within each block are causal, on the grounds that within the Atlantic population, LD decays very rapidly, over a hundred bp or so. This would be very interesting, implying that complex adaptive alleles build up as a result of successive substitutions in the same region. However, careful arguments are needed to exclude the more obvious alternative, that selection has raised specific haplotypes to high frequency. Suppose that the original population was indeed at linkage equilibrium. Under selection in a new environment, many alleles might increase at the same locus – either from standing variation or from new mutations. Such "soft sweeps" might not be detected in these data. However, at a subset of loci, one or a few haplotypes might increase, and their size would reflect the time since adaptation – presumably a few thousand generations, corresponding to perhaps 10-100kb. Even if two or three successive causal alleles were involved, the great majority of SNPs would still be neutral. Seeing LE within the base population does not address this issue; and the ascertainment bias means that only more or less hard sweeps can be detected.

Paragraph six subsection “Genetic adaptation to a new niche environment”: There needs to be a proper test of whether it is surprising that significant variants cover genes in certain functional classes – I have no idea how many genes may be involved in osmoregulation, for example. This is especially an issue because haplotype blocks may cover multiple genes. A proper permutation test is needed here.

Subsection “Genomic distribution of causal variants”: The section on genomic distribution of causal variants was confused. It should be possible to make a rigorous statistical estimate of the fraction of markers that are likely to be causal from the extent of enrichment, but this needs to be done carefully.

*Reviewer #3:*

This is a comprehensive and nice study and well-written paper, with possible implications for inference on ecological adaptation of species other than the Atlantic herring.

My major comment is that I was surprised by how the paper was written with respect to the literature. After reading Lamichhaney et al. 2012 from the same group it is clear that the current study is a kind of (nice) follow-up from that. Yet the authors only refer to Lamichhaney et al. 2012 once in the Introduction and once in the Results and not at all in the Discussion. In my opinion, the authors should be upfront of what they found in L2012, how this study goes well beyond that, and how the current results confirms or contradicts previously reported results.

My quick reading of Lamichhaney et al. 2012 is that the authors reported: (i) low level population differentiation, (ii) evidence for local adaptation (salinity), (iii) evidence for selection on haplotype blocks, (iv) candidate genes under natural selection (salinity) and (v) evidence of natural selection of reproduction (spawning season).

[Editors’ note: what now follows is the decision letter after the authors submitted for further consideration.]

Thank you for submitting your work entitled "The genetic basis for ecological adaptation of the Atlantic herring revealed by genome sequencing" for consideration by *eLife*. Your article has been reviewed by three peer reviewers, including Nick Barton and Magnus Nordborg, who is a member of our Board of Reviewing Editors, and the evaluation has been overseen Diethard Tautz as the Senior Editor.

The reviewers have discussed the reviews with one another and the Reviewing Editor has drafted this decision to help you prepare a revised submission.

Summary:

As emphasized in the decision letter encouraging you to resubmit this manuscript, the reviewers unanimously agree that this is one of the best data sets in existence for addressing questions about the architecture of local adaptation. The resubmitted version is greatly improved, but is still somewhat frustrating in that it was felt that much more could be done with the data. However, it was recognized that this would take the analysis well beyond the current state of the art, and that it would not be constructive to demand this. We actually do not yet have the intellectual framework for thinking about these kinds of data. It says on page 5 that "the results provide a comprehensive and detailed view on the genetic architecture underlying ecological adaptation". So what is the answer? How far can we go without reciprocal transplant studies?

Better then to publish, and be clear about what has and has not been demonstrated.

Essential revisions:

The distinction between causal and non-causal SNPs needs to be more clearly spelled out. The paper identifies a very large number of SNPs that differ significantly between groups. These tend to be clustered on the genome, and also enriched for certain functional types. Some clusters may arise by chance, whilst others may include one or more causal alleles, which tend to be in candidate loci. The enrichment analysis shows the clearest evidence for selection, but it seems to us that this could be consistent with multiple causal alleles at ~ 20 loci, rather than the "thousands" of SNPs cited in the abstract. It should be possible to estimate the minimum number of causal loci consistent with the observed clustering and enrichment, at least roughly.

The existence of long haplotypes distinguishing the different population is emphasized, but it is not clear whether these are in any sense unusual given the low effective population (surprisingly low given the vast census sizes in this species). It should be possible to use standard coalescent simulation to at least test whether the observed haplotype lengths are long compared to neutral expectations under the estimated demography and a range of plausible values for the recombination rate.

---

## [Author Response]

[Editors’ note: the author responses to the first round of peer review follow.]

*The consensus opinion of the reviewers is that this is a really interesting study in an excellent study system, but that the analysis is insufficient and several of the conclusions not supported. In particular, the reviewers were not convinced that you can actually identify the causal sites, and several conclusions rest on this. However, exploring the limits of what may be concluded is also of great interest, especially given the quality of the data. In essence, the analysis does not do the data justice.*

First we would like to thank the reviewers for their constructive criticism of our paper that has clearly helped us to improve the paper. The most important change in the revised version is that we have included a much improved analysis of the genomic distribution of causal changes and this shows that we had underestimated the role of changes in the coding sequence based on the previous analysis.

Reviewer #1:

*This is a substantial piece of work addressing an important question using a beautiful and promising system. It is certainly the case that identifying loci that show evidence having been under selection is much easier in gigantic population with little population structure, as appears to be the case for herring. Unfortunately, the paper suffers from several conceptual/logical flaws and I do not think the major conclusions are supported. The confusion begins in the first sentence of the abstract, where it is stated that "Ecological adaptation is of major relevance…, but the underlying genetic factors are typically hard to study in natural populations due to confounding between population structure and signatures of selection". It is indeed true that population structure makes it hard to identify signatures of selection, but who says that genes involved in ecological adaptation will exhibit such signatures? Generally, such signals arise as a consequence of strong selection on individual genes. For a quantitative trait, is not clear that there will be any such genes, and is also not clear that any such genes found will explain much of the variation.*

We have changed the sentence in the Abstract to “…due to confounding between genetic differentiation caused by natural selection and by genetic drift in subdivided populations” to make a more specific statement. Furthermore, we think this is a major merit of the paper that it demonstrates that very convincing evidence of natural selection can be detected in this system.

*Furthermore, there is no phenotype in the present study, nor any heritability. Thus the comparison with human height (Discussion section) does not make sense. You have no idea how much of the variation for fitness your SNPs explain. Indeed it is formally possibly that all your SNPs are associated with defense against a parasite with a life-cycle that depends on salinity. I obviously don't believe this, but the point remains that in order to discuss the genetic architecture of adaptation, there needs to be genetics in the study. For example, a reciprocal transplant study that quantified the fitness effects of the identified loci.*

We disagree with the statement that there is no phenotype or genetics in this study. We have sampled fish from the same locality spawning in May, July and September and these were phenotypically classified as spring-, summer- and autumn-spawners. Our sequence analysis revealed highly significant allele frequency differences at hundreds of loci between these groups. This is a genetic result! However, the reviewer is 100% correct that we cannot estimate heritabilities since we do not have genotype-phenotype relationships for individual fish. We have therefore revised the Discussion and now explain that we cannot determine how much of the phenotypic variation these loci control.

Speaking of associations, I don't understand how the threshold for calling frequency differences significant was set (Figure 1—figure supplement 1). This is not explained ("used a QQ-plot" is not an explanation).

We have decided to rather use a Bonferroni correction since that is a conservative threshold in a study like this. This means that the significance threshold has changed from p=10x10^-20^ to 10x10^-10^, but this did not have any major effects on the main conclusions in this study. We still keep the Q-Q plot as Figure 1—figure supplement 1 to demonstrate that the distribution of P-values in the association deviates dramatically from the expected distribution under the null hypothesis of no genetic differentiation.

*It is also stated that the presence of haplotype blocks was striking. Why? Presumably these are simply the consequence of selection? Are they longer than you would expect? Why? What is the evidence that they contain multiple causal sites as opposed to being the result of linkage drag?*

We agree that we cannot exclude the possibility that linkage drag is an important explanation for haplotype blocks so we have modified the text in the Discussion. However, we do think that the new data showing the presence of multiple non-synonymous substitutions in many of the genes showing the strongest genetic differentiation support the hypothesis of haplotype evolution involving several causal changes within the haplotype blocks.

*Regardless of why they are there, the existence of these blocks mandate extreme caution when trying to decide which sites are causal. It is by now clear that local haplotype structure, especially when coupled with allelic heterogeneity causes large number of local spurious associations, and that, as a consequence, the most strongly associated SNP is NOT likely to be the causal one: it is likely to be an intermediate-frequency SNP that happens to tag the underlying variants well… A consequence of this is that most of the conclusions about what kinds of SNPs do what (Subsection “Genomic distribution of causal variants”*

*) are not supported. You find few significantly associated non-synonymous substitutions, but this could simply be because they are all rare, and can only be found by proxy. It does make sense to compare different kinds of regions, as you do, but everything has to be controlled for allele frequency.*

This section has been completely revised and the text rewritten. In fact our new improved analysis gives a different and more correct picture about what types of genetic changes are underlying ecological adaptation in herring.

*On a more population genetics level, I was struck by your low nucleotide diversity coupled with a rapid decay of linkage disequilibrium. Unless mutation- and recombination rates are very different from other organisms, they make no sense. My guess is that your nucleotide diversity is underestimated because you are detecting heterozygotes in single individuals, and that your rate of linkage disequilibrium decay is overestimated because of bad SNPs.*

The nucleotide diversity is calculated in both the ~70x coverage reference individual as well as the 16 ~10x coverage individual samples with very consistent results, indicating that the diversity is not substantially underestimated due to lack of power to detect heterozygous sites. Also, while perhaps low in some contexts, the observed value is higher than for other species of fish (See Table 1).

Regarding the genotype quality and LD decay, we observe strong concordance between SNPs called from sequencing and SNP chip (in individuals ~90% of heterozygote sites have matching calls (data not shown), and see Supplementary Figure 3 for the correlation between allele frequency in pooled samples and SNP-chip data). Admittedly, the SNP-chip markers are not a random subset of all SNPs and are likely to be of better quality than the genomic average, but we still have reasons to believe that our genotypes are generally good; we have also used the SNP chip for some pedigree analysis (data not shown). Also, inaccurate genotype calls are not likely to correlate with physical distance between SNPs, and will therefore not affect the shape of the LD decay curve. So, perhaps mutation and/or recombination data differ from other organisms?

*Third paragraph subsection “Genetic basis underlying timing of reproduction” This something not right about this argument. Why would rare SNPs affect nucleotide diversity one way or another?*

The text has been revised. The message is that spring-spawning herring possess a larger number of variable sites but the average heterozygosity per site is lower.

*Subsection “Genomic distribution of causal variants”: "Some of these will be false positives due to close linkage to true positives but many will be true positives" I discuss this above, but it would be important to realise that, without assumptions about the meaning of "some" and "many", it is not possible to conclude any of what you conclude further down.*

This entire section has been revised and rewritten.

*Same section: "Enrichment" compared to what?*

The section has been rewritten and a new Figure 5 has been prepared. Now it is better explained that we see an enrichment compared with the genome-wide frequency.

Reviewer #2:

*This paper uses whole-genome sequencing from 20 population pools, supported by SNP genotyping of ~360 individuals, to identify variants associated with differences between Baltic and Atlantic, and between autumn and spring spawning herring. This is an impressive set of data, and as the authors argue, the recent divergence, large population size and low differentiation makes herring exceptionally well suited to identifying genes responsible for adaptation. Thus, the study could in principle be a very nice contribution to eLife. However, the interpretation of the results seems quite naive: more detailed and rigorous argument is needed to be convincing. SNP and structural variants associated with divergence are clustered into haplotype blocks that may span multiple genes, and up to 200kb. It seems to be assumed that all (or at least, many) of the variants within each block are causal, on the grounds that within the Atlantic population, LD decays very rapidly, over a hundred bp or so. This would be very interesting, implying that complex adaptive alleles build up as a result of successive substitutions in the same region. However, careful arguments are needed to exclude the more obvious alternative, that selection has raised specific haplotypes to high frequency. Suppose that the original population was indeed at linkage equilibrium. Under selection in a new environment, many alleles might increase at the same locus – either from standing variation or from new mutations. Such "soft sweeps" might not be detected in these data. However, at a subset of loci, one or a few haplotypes might increase, and their size would reflect the time since adaptation – presumably a few thousand generations, corresponding to perhaps 10-100kb. Even if two or three successive causal alleles were involved, the great majority of SNPs would still be neutral. Seeing LE within the base population does not address this issue; and the ascertainment bias means that only more or less hard sweeps can be detected.*

For sure, we do not think that all variants in a haplotype block is causal, in fact we assume that the majority of highly differentiated SNPs are linked to causal SNPs. However, we think it is unlikely that these fairly large haplotype blocks under selection are due to single causal variants but we agree that it is difficult to exclude this possibility and we have therefore softened the discussion of this issue.

*Paragraph six subsection “Genetic adaptation to a new niche environment”: There needs to be a proper test of whether it is surprising that significant variants cover genes in certain functional classes – I have no idea how many genes may be involved in osmoregulation, for example. This is especially an issue because haplotype blocks may cover multiple genes. A proper permutation test is needed here.*

We have attempted to carry out proper GO analysis to address this but our conclusion is that the GO annotations of fish genes are too poor to make this a powerful analysis. However, we find it encouraging that several of the top hits make perfect sense, such as the prolactin receptor and high choriolytic enzyme genes associated with adaptation to Baltic Sea – Atlantic Ocean that differs dramatically as regards salinity, and the *TSHR, SOX11, CALM* and *ESRB* loci associated with spawning time which all have an established role in reproductive biology.

*Subsection “Genomic distribution of causal variants”: The section on genomic distribution of causal variants was confused. It should be possible to make a rigorous statistical estimate of the fraction of markers that are likely to be causal from the extent of enrichment, but this needs to be done carefully.*

This section has been completely revised and the text rewritten.

Reviewer #3:

*This is a comprehensive and nice study and well-written paper, with possible implications for inference on ecological adaptation of species other than the Atlantic herring. My major comment is that I was surprised by how the paper was written with respect to the literature. After reading Lamichhaney et al. 2012 from the same group it is clear that the current study is a kind of (nice) follow-up from that. Yet the authors only refer to Lamichhaney et al. 2012 once in the Introduction and once in the Results and not at all in the Discussion. In my opinion, the authors should be upfront of what they found in Lamichhaney et al. 2012, how this study goes well beyond that, and how the current results confirms or contradicts previously reported results. My quick reading of Lamichhaney et al. 2012, is that the authors reported: (i) low level population differentiation, (ii) evidence for local adaptation (salinity), (iii) evidence for selection on haplotype blocks, (iv) candidate genes under natural selection (salinity) and (v) evidence of natural selection of reproduction (spawning season).*

We have revised the Introduction and now cite our previous work more extensively in the Introduction.

[Editors’ note: the author responses to the re-review follow.]

*Summary: As emphasized in the decision letter encouraging you to resubmit this manuscript, the reviewers unanimously agree that this is one of the best data sets in existence for addressing questions about the architecture of local adaptation. The resubmitted version is greatly improved, but is still somewhat frustrating in that it was felt that much more could be done with the data. However, it was recognized that this would take the analysis well beyond the current state of the art, and that it would not be constructive to demand this. We actually do not yet have the intellectual framework for thinking about these kinds of data. It says on page 5 that "the results provide a comprehensive and detailed view on the genetic architecture underlying ecological adaptation". So what is the answer? How far can we go without reciprocal transplant studies?*

We agree that we really do not know how much of the genetic variance our loci control so we have changed this sentence as follows:

“The results provide a comprehensive list of hundreds of independent loci underlying ecological adaptation and shed light on the relative importance of coding and non-coding variation.”

*Better then to publish, and be clear about what has and has not been demonstrated. Essential revisions: The distinction between causal and non-causal SNPs needs to be more clearly spelled out. The paper identifies a very large number of SNPs that differ significantly between groups. These tend to be clustered on the genome, and also enriched for certain functional types. Some clusters may arise by chance, whilst others may include one or more causal alleles, which tend to be in candidate loci. The enrichment analysis shows the clearest evidence for selection, but it seems to us that this could be consistent with multiple causal alleles at ~ 20 loci, rather than the "thousands" of SNPs cited in the abstract. It should be possible to estimate the minimum number of causal loci consistent with the observed clustering and enrichment, at least roughly.*

In the current version of the paper we provide conservative estimates of the minimum number of independent loci detected in this study. We requested a statistical support of at least 1x10^-20^ to consider a region to make sure that the number of false positives is very low if any. We took into account gaps in the assembly, we looked at the correlation between SNP P-values within regions and we required a distance of at least 20 kb without highly significant SNPs to close a region. Using these criteria we identified 472 independent loci associated with the Atlantic-Baltic contrast and 125 independent loci distinguishing autumn- and spring spawners. It is clear from Figure 3 and Figure 4 that the number of loci cannot be as few as ~20.

*The existence of long haplotypes distinguishing the different population is emphasized, but it is not clear whether these are in any sense unusual given the low effective population (surprisingly low given the vast census sizes in this species). It should be possible to use standard coalescent simulation to at least test whether the observed haplotype lengths are long compared to neutral expectations under the estimated demography and a range of plausible values for the recombination rate.*

We think the most important question to explore is if the large haplotype blocks are caused by strong, fairly recent selective sweeps or the evolution of haplotypes harbouring multiple causal alleles. We took advantage of the fact that these two models give entirely different predictions as regards the nucleotide diversity within the differentiated regions within populations. The selective sweep/hitchhiking model is expected to lead to a loss of nucleotide diversity whereas the haplotype evolution model predicts that nucleotide diversity in the differentiated regions can be as high or even higher than in the rest of the genome if these haplotype blocks have been maintained by balancing selection. Our analysis presented in the revised section “Adaptive haplotype blocks are maintained by selection” and in the new Figure 5 shows that nucleotide diversity is in fact higher in the regions showing strong differentiation even within populations. We therefore conclude that our data are consistent with evolution of adaptive haplotypes carrying multiple causal variants. We think this is an important improvement of the paper.